# Anion-exchange-mediated internal electric field for boosting photogenerated carrier separation and utilization

Tong Han[1,6], Xing Cao [1,6], Kaian Sun[1,6], Qing Peng[1✉], Chenliang Ye[1], Aijian Huang[1], Weng-Chon Cheong [2], Zheng Chen [3], Rui Lin[4], Di Zhao[5], Xin Tan [1], Zewen Zhuang [1], Chen Chen[1✉], Dingsheng Wang [1] & Yadong Li [1✉]

Heterojunctions modulated internal electric field (IEF) usually result in suboptimal efficiencies in carrier separation and utilization because of the narrow IEF distribution and long migration paths of photocarriers. In this work, we report distinctive bismuth oxyhydroxide compound nanorods (denoted as BOH NRs) featuring surface-exposed open channels and a simple chemical composition; by simply modifying the bulk anion layers to overcome the limitations of heterojunctions, the bulk IEF could be readily modulated. Benefiting from the unique crystal structure and the localization of valence electrons, the bulk IEF intensity increases with the atomic number of introduced halide anions. Therefore, A low exchange ratio (~10%) with halide anions ($I^-$, $Br^-$, $Cl^-$) gives rise to a prominent elevation in carrier separation efficiency and better photocatalytic performance for benzylamine coupling oxidation. Here, our work offers new insights into the design and optimization of semiconductor photocatalysts.

[1] Department of Chemistry, Tsinghua University, Beijing, China. [2] Department of Physics and Chemistry, Faculty of Science and Technology, University of Macau, Macao SAR, China. [3] College of Chemistry and Materials Science, Anhui Normal University, Wuhu, China. [4] Nanoinstitute Munich, Ludwig-Maximilians-Universität München, Munich, Germany. [5] Key Laboratory of Cluster Science, Ministry of Education of China, Beijing Key Laboratory of Photoelectronic/Electrophotonic Conversion Materials, School of Chemistry and Chemical Engineering, Beijing Institute of Technology, Beijing, China. [6] These authors contributed equally: Tong Han, Xing Cao, Kaian Sun. ✉email: pengqing@mail.tsinghua.edu.cn; cchen@mail.tsinghua.edu.cn; ydli@mail.tsinghua.edu.cn

In recent years, light-powered catalytic organic synthesis has garnered increasing research interests. During photocatalysis, the separation of electron-hole pairs within the semiconductor catalysts, combined with the following energy transfer processes, can readily generate a large amount of highly active species (such as radicals and singlet oxygen[1-4]. Notably, these active species could, under green and mild conditions, participate in a variety of organic reactions, including hydrogenation[5,6], epoxidation[7,8], alcohol oxidation[9,10], selective oxidation of aromatic compounds[11,12], and even some reactions that are rather challenging in thermal catalysis. Yet still, currently for heterogeneous photocatalysts, there exist the common issues of rapid recombination of photocarriers and the resulting low efficiency of carrier separation and utilization, which would hamper the high-performance catalysis of organic reactions, and thus their applications have so far been limited primarily to environment-related aspects such as degradation of organics, air purification and water photolysis[13-17]. A strategy extensively adopted to boost the carrier separation efficiency is to construct composite materials featuring heterojunctions, in which the internal electric field (IEF), resulting from the different band structures at the interface, is expected to facilitate the carrier separation and migration[18,19].

However, owing to the poor lattice match at the interface and the resulting structural defects, the composite materials generally suffer from low structural stability and constrained migration of photocarriers; moreover, the IEF thus generated locates merely at the interface, with a rather limited distribution, and therefore is not likely competent to solve the challenging problem of bulk carrier separation[20-24]. It can thus be envisioned that, if we could somehow synthetically modify the structure of photocatalysts to build a more intense IEF within the entire materials (rather than merely at the heterojunction interfaces) while preserving the original high crystallinity, and to accelerate the carrier migration to substrate molecules, the resulting photocatalysts would feature a high efficiency of spatial separation of carriers, as well as a high utilization of photogenerated charges[25-27].

Following the above discussion, in this work we designed and prepared Sillenite-structured single-crystalline bismuth oxyhydroxide compound nanorods (denoted as BOH NRs) with high aspect ratios (1–6 μm in length, 10–30 nm in diameter). Each NR is composed of multiple (10–30) alternating layers of $[Bi_2O_2]^{2+}$ cations and $OH^-$ counteranions stacked along a specific crystallographic direction perpendicular to the longitudinal axis (Fig. 1). Benefiting from this unique structure, the nanorods feature open

channels exposed at the surface, and internal electric fields between the alternating layers. Upon light illumination, the generated carriers would be readily separated and transported to the surface along the direction perpendicular to the longitudinal axis of the nanorods, which not only achieves the carrier separation in the bulk phase, but also significantly shortens the migration paths for carriers, and thus effectively promotes the utilization of photogenerated electrons and holes. More importantly, this unique structure features an open and ordered arrangement, which enables the modulation of IEF intensity by simply modifying the anion layer; therefore, this catalyst constitutes an ideal model for the theoretical and experimental studies on the relationship between the compositions, structures, and IEF of photocatalysts and their catalytic performance. Furthermore, the efficiency of carrier separation and transportation, as well as the photocatalytic activity for benzylamine oxidation, are improved to a similar extent, which reveals that the built-in electric field is the inherent driving force for carrier separation, transportation, and utilization.

Density functional theory (DFT) calculations reveal that, after halide exchange, the bulk IEF within the BOH NRs becomes significantly intensified; the IEF intensity increases with the atomic number of halide anions, and a similar trend was found for the benzylamine conversion over different halide-exchanged catalysts. Such a difference in the efficiency of carrier separation and utilization results simply from the different halide anions introduced, and could be attributed primarily to the variation in the electrostatic potential difference between neighboring layers (i.e., the IEF), which is sensitive to the localization of valence electrons and the interlayer spacing. Our study here not only develops a high-performance photocatalyst featuring bulk-IEF-facilitated charge separation, but also clearly demonstrates the effectiveness of boosting IEF intensity and photocatalytic performances via ion exchange, and therefore offers new insights for exploring advanced photocatalysts and high-performance photocatalytic organic reactions.

## Results

**Synthesis strategy and characterization of BOH nanorods**. The Sillenite-structured BOH NRs were prepared by a hydrothermal method, with surfactants added to regulate the hydrolysis of Bi$(NO_3)_3$ precursor. Specifically, Bi$(NO_3)_3 \cdot 5H_2O$, PVP (polyvinyl pyrrolidone-8K), and mannitol were dissolved in deionized water under stirring, and the resulting mixture was subjected to

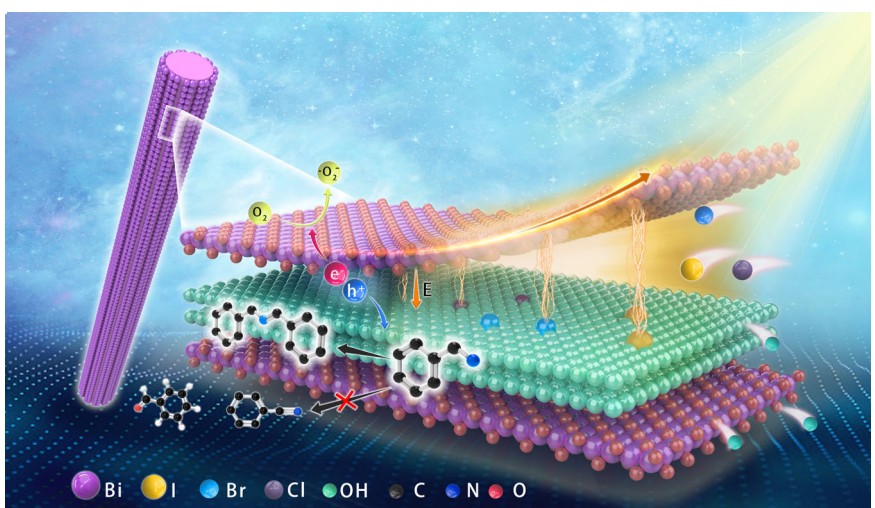

**Fig. 1 Schematic illustration of IEF intensity change induced by ion exchange.** The IEF intensity is strengthened for BOH nanorods using halide anions exchange, and the holes accumulated at the surface-exposed anion layers can be utilized to oxidize benzylamine.

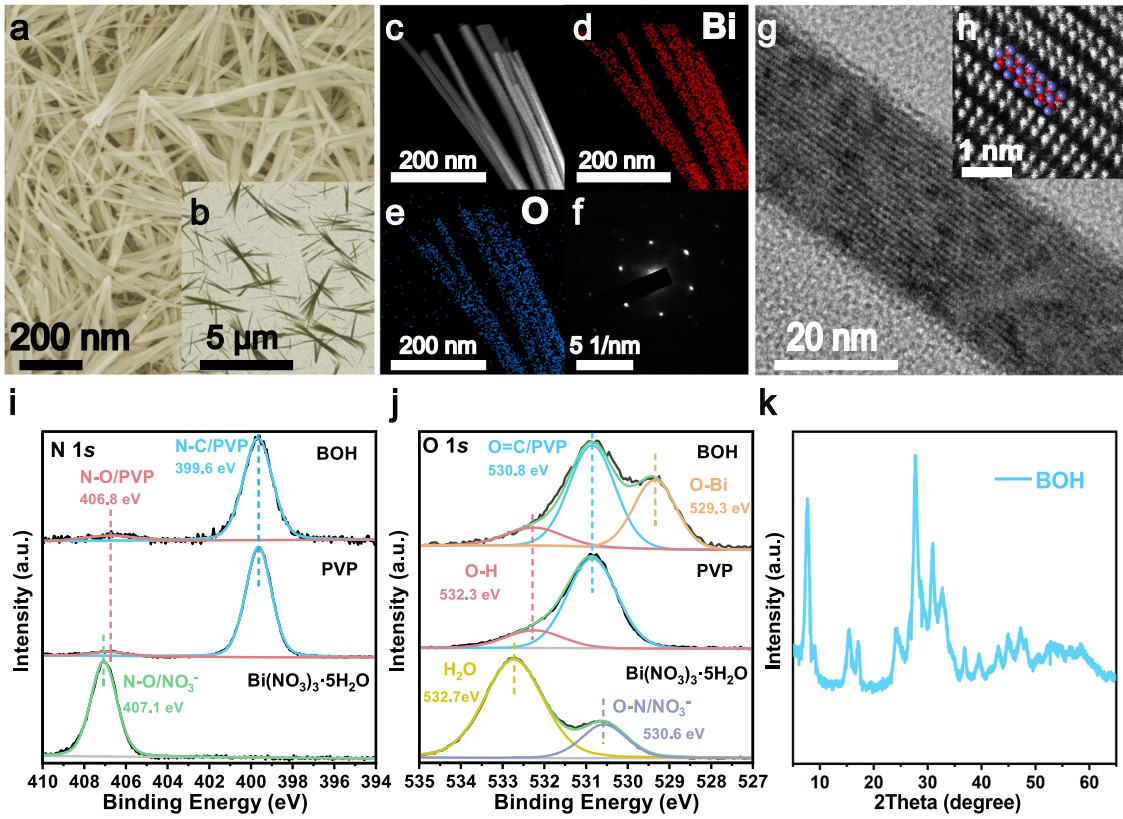

**Fig. 2 Structure characterizations of BOH. a** SEM image. **b** TEM image. **c** HAADF-STEM image. **d**, **e** EDS mapping for Bi and O, respectively. **f** SAED pattern. **g** HRTEM image. **h** Atomic-resolution AC HAADF-STEM image. And the overlaying model image showing the structure of $[Bi_2O_2]^{2+}$ layer in $Bi_2O_2(OH)(NO_3)$ (red balls; Bi atoms; blue balls, O atoms). **i**, **j** XPS spectra for N, O in BOH, PVP, and $Bi(NO_3)_3 \cdot 5H_2O$, respectively. **k** XRD pattern.

hydrothermal treatment at 160 °C for 24 h. As shown in the scanning electron microscopy (SEM) and transmission electron microscopy (TEM) images (Fig. 2a and the inset Fig. 2b), the product BOH has a morphology of nanorods with high aspect ratios (10–30 nm in diameter, 1–6 μm in length); most NRs aggregate into bundles by connecting at the middle or one end. The introduction of PVP and mannitol proved critical for the formation of NRs: as shown in Supplementary Fig. 1, without mannitol, the product has a rod-like morphology, but the NRs are shorter in length and not uniform in diameter; without PVP, the product has a sheet-like morphology. These results indicate that PVP can regulate the growth of BOH along with specific directions, thus exposing specific facets and leading to the formation of NRs; mannitol can promote the dissolution of $Bi(NO_3)_3 \cdot 5H_2O$ in water, leading to uniform nucleation of BOH and the resulting NRs with high uniformity and aspect ratios[28].

The high-angle annular dark-field scanning transmission electron microscopy (HAADF-STEM) image and the energy-dispersive X-ray (EDX) mapping results (Fig. 2c–e) confirm that in the BOH NRs, the Bi and O elements are evenly distributed. The selected-area electron diffraction (SAED) pattern (Fig. 2f) and high-resolution TEM (HRTEM) image (Fig. 2g) reveal that each BOH NR is of single crystallinity and has well-ordered lattice fringes. The hydrolysis of $Bi(NO_3)_3 \cdot 5H_2O$ can yield multiple products including $Bi_5O_7(NO_3)$ or basic bismuth nitrates ($Bi_6O_6(OH)_2(NO_3)_4 \cdot 2H_2O$, $Bi_6O_5(OH)_3(NO_3)_5 \cdot 3H_2O$, $Bi_2O_2(OH)$ $(NO_3)$, etc.). All of these products are made of backbones of [Bi–O] layers, with anion layers intercalated in between; the main differences are the arrangement of Bi and O atoms in [Bi–O] layers, and the types and amounts of anions (such as $NO_3^-$ and $OH^-$)[29–31]. We performed aberration-corrected HAADF-STEM (AC HAADF-STEM) with atomic resolution. Supplementary Fig. 2

shows that the BOH NR is composed of ~10 ordered layers arranged in parallel, each layer with a thickness of ~1 nm. In Fig. 2h, it can be clearly observed that each layer has two arrays of Bi atoms (Bi atoms appear as bright dots in the image, whereas O atoms are barely observable owing to the small atomic number), and the inter-array spacing is 0.27 nm, which is identical to that within the $[Bi_2O_2]^{2+}$ layer of the known compound $Bi_2O_2(OH)$ $(NO_3)$ (that is, the distance between two neighboring red balls in Fig. 2h) (for the two-dimensional structure of $Bi_2O_2(OH)(NO_3)$, see Supplementary Fig. 3). The results above confirm that our BOH NRs have a layered $[Bi_2O_2]^{2+}$ structure similar to that in $Bi_2O_2(OH)(NO_3)$, and the interlayer channels are openly exposed. The alternating $[Bi_2O_2]^{2+}$ and anion layers are stacked via van der Waals interaction, forming a layered Sillenite structure; between the neighboring $[Bi_2O_2]^{2+}$ and anion layers exists a perpendicular IEF, which could facilitate the carrier separation[32,33]. In addition, the IEF is also perpendicular to the normal of the exposed facets of BOH, and thus shortens the migration path for photocarriers, which is conducive to the transport and separation of charges[34].

In order to probe the chemical identity of the intercalating anions, we performed X-ray photoelectron spectroscopy on the $Bi(NO_3)_3 \cdot 5H_2O$ precursor, the PVP surfactant, and the product BOH NRs. The N 1s spectra (Fig. 2i) show that the binding energies of N (399.6 eV and 406.8 eV) in the BOH sample are similar to those for PVP, and different with that for $NO_3^-$ (407.1 eV). These results indicate that there barely exists any $NO_3^-$ within the 10 nm subsurface region (which constitutes almost the entire volume) of the BOH NR. Figure 2j shows the O 1s peaks at 529.3 eV, 530.8 eV and 532.3 eV for BOH, corresponding to the binding energies of O atoms in Bi–O bonds in the $[Bi_2O_2]^{2+}$ layer, the C=O groups in PVP, and $OH^-$ anions, respectively. Therefore, we infer that the BOH NRs have a backbone structure

similar to that for $Bi_2O_2(OH)(NO_3)$, only with $OH^-$ anions intercalated between the $[Bi_2O_2]^{2+}$ layers. For our BOH NRs, the XRD pattern (Fig. 2k) does not match well with any standard XRD pattern available, including any hydrolysis products of $Bi(NO_3)_3$ such as $Bi_2O_2(OH)(NO_3)$ (Supplementary Fig. 4). By comparing with the XRD pattern for the hydrothermal product without PVP (denoted as BOH-nPVP) (Supplementary Fig. 5), we confirmed that PVP only plays a role in regulating the nucleation and growth of products. Considering the common hydrolysis products of $Bi(NO_3)_3$, it could be inferred that our BOH has a similar basic structure with alternating [Bi–O] and anion layers. Therefore, we simulated the XRD pattern of $Bi_2O_2(OH)_2$ was fitted according to the hypothesis (Supplementary Fig. 6), which fitted well to the experimental results. BOH has a larger cation and anion layer spacing than $Bi_2O_2(OH)(NO_3)$, which could be interpreted that: $NO_3^-$ anion has a larger radius (2.00 Å) than $OH^-$ (0.89 Å), so when $OH^-$ anions are intercalated between the $[Bi_2O_2]^{2+}$ layers (in the case of BOH) instead of $NO_3^-$ (in the case of $Bi_2O_2(OH)(NO_3)$), the bridging effect of the anions would become less pronounced, leading to increased spacing between neighboring $[Bi_2O_2]^{2+}$ layers owing to coulombic repulsion[35]. Moreover, the altered symmetry for BOH also results in diffraction peaks located at different angles from those for $Bi_2O_2(OH)(NO_3)$ and more diffraction peaks.

Owing to the alternating arrangement of $[Bi_2O_2]^{2+}$ and anion layers, the hydrolysis products of $Bi(NO_3)_3 \cdot 5H_2O$ are usually $Bi_5O_7(NO_3)$ or basic bismuth nitrates with sheet-like morphologies; yet in this work, by introducing PVP and mannitol during the hydrolysis, we obtained rod-like structures with barely any $NO_3^-$ and with interlayer channels openly exposed. Compared with conventional sheet-like products, this rod-like structure could not only significantly increase the specific surface area (for example, compared with BOH-nPVP, the specific surface area of BOH nanorods is increased from $11.7\ m^2/g$ to $31.7\ m^2/g$ (Supplementary Fig. 7)), but also shorten the paths for photocarriers to migrate from the interior to the surface, and the holes accumulated at the surface-exposed anion layers can be utilized to activate substrate molecules. In addition, by taking advantage of the openly exposed channels, the anion layers may be modified in both composition and structure.

**Synthesis strategy and characterization of BOX nanorods (X = Cl, Br, I).** We used the BOH NRs as precursors for subsequent anion exchange experiments. Specifically, BOH NRs were dispersed in deionized water, and a proper amount of KX (X = Cl, Br, I) was added. The mixture was sonicated, sealed in an autoclave, and then heated at 60 °C for 12 h. As shown in Fig. 3a, during the reaction process, the $OH^-$ anions between the $[Bi_2O_2]^{2+}$ layers could partially exchange with halide anions, and the resulting products could well preserve the rod-like morphology and the backbone structure of unmodified BOH NRs. The products after the exchange with KI, KBr, and KCl are hereafter denoted as BOH-I, BOH-Br, and BOH-Cl, respectively (and collectively as BOH-X).

Similar to the pristine BOH NRs, all three BOH-X samples have a rod-like morphology with diameters of 10–30 nm and lengths of 1–6 μm (Fig. 3b–d and Supplementary Fig. 8), and the introduction of halogen does not change the specific surface area of the material (Supplementary Fig. 7). EDX mapping (Fig. 3e–g and Supplementary Figs. 9–11) showed uniform distributions of halogen atoms over the NRs after anion exchange; HRTEM-STEM images and the corresponding SAED patterns (Fig. 3h–j) revealed well-preserved single crystallinity for the BOH-X NRs. The XRD patterns for the three BOH-X samples (Supplementary Fig. 12) are almost identical; yet the primary peaks (at 2 *Theta* =

7.7° for BOH) shift toward lower angles to different extents, indicating that the intercalation of halide anions has altered the interlayer spacings. In addition, the XPS spectra (Supplementary Fig. 13) revealed that the binding energies of Bi 4*f* in the BOH-X samples were elevated by less than 0.2 eV with respect to that for BOH, confirming that the halide anions had been introduced successfully. By comparing peak areas of the Bi–O bond and O–H bond in the XPS spectra (Supplementary Fig. 14), we found that the ratios of the O–H bond became lower for BOH-X, again confirming the success of halide exchange and intercalation. Semi-quantitative analyses based on the XPS data revealed that after anion exchange, the $X^-/Bi^{3+}$ ratios are 0.12 (for I), 0.09 (for Br), and 0.11 (for Cl), indicating similar activities of ion exchange for the halide anions. With more halogen introduced (2–20 times), the TEM and XRD patterns of the ion-exchange products did not change significantly (Supplementary Figs. 15–18, and, unless otherwise stated, BOH-X samples refer to ion exchange products with 0.1 mmol halogenated potassium added). The XPS semi-quantitative results showed that the maximum $I^-$ introduction amount only reached to 17% (Supplementary Fig. 19), indicating the limited exchange capacity of halide ions. To unveil the distribution of halogen atoms, we selected BOH-I as a representative and performed High-resolution XPS experiments with $Ar^+$ sputtering at different depths (Fig. 4a, b and Supplementary Fig. 20). As the sputtering depth (14 nm) is nearly equal to (or larger than) the radii of the NRs, the results suggest that $I^-$ anions are evenly distributed within the entire volume, rather than merely at the surface. In addition, no prominent peaks corresponding to N were observed at the subsurface region, again implying that the N atoms come of PVP, and are distributed primarily at the surface. All the above results confirmed the efficacy of modifying the bulk anion layers of BOH via hydrothermal halide-anion exchange.

The introduction of halide anions may modify the band structure of the photocatalyst. As shown in the UV–vis diffuse reflectance spectra (DRS) (Fig. 4c and Tauc plot in Supplementary Fig. 21), the exchange with $Cl^-$ and $Br^-$ do not affect the absorption edge for the pristine BOH (about 344 nm), indicating that BOH, BOH-Cl, and BOH-Br have similar light absorption ranges and similar bandgaps (3.54–3.60 eV). By contrast, the exchange with $I^-$ induce a redshift of the absorption edge from ~340 nm to ~400 nm, which means a narrower bandgap for BOH-I (3.13 eV) and enhanced absorption in the visible region. The positive slopes in the Mott–Schottky plots (Supplementary Fig. 22) reveal a character of n-type semiconductor for all four samples, and the flat band potentials ranging within 0.02–0.15 eV. Combined with the XPS valence band spectra of samples (Supplementary Fig. 23), the energy band diagram with respect to the normal hydrogen electrode (Supplementary Fig. 24, see SI for calculation details) is obtained. All BOH-X samples have a similar conduction band minimum (CBM) as BOH (−1.83– −1.93 eV). The Valence Band Maxime (VBM) of BOH are similar to that of BOH-Cl and BOH-Br (about 1.70 eV), however, BOH-I have the highest VBM (1.24 eV). To sum up, the anion exchange induces a moderate change in VMB only for BOH-I, whereas its influences on the band structures of BOH-X are rather limited.

To explore the influence of the halogens on the internal electric field, the IEF intensity was measured based on the model proposed by Kanata et al.[36,37]. (See SI for details). According to the model, the IEF intensity is determined by the surface voltage ($V_s$) and the surface charge density ($\rho$). Compared with BOH, all BOH-X exhibit the more vigorous surface photovoltage response ((Supplementary Fig. 25) and higher photocurrent density (Supplementary Fig. 26). Particularly, the surface photovoltage intensity and photocurrent density of BOH-I are 1.9 times and 2.1 times as high as that of BOH, respectively. It can be found that

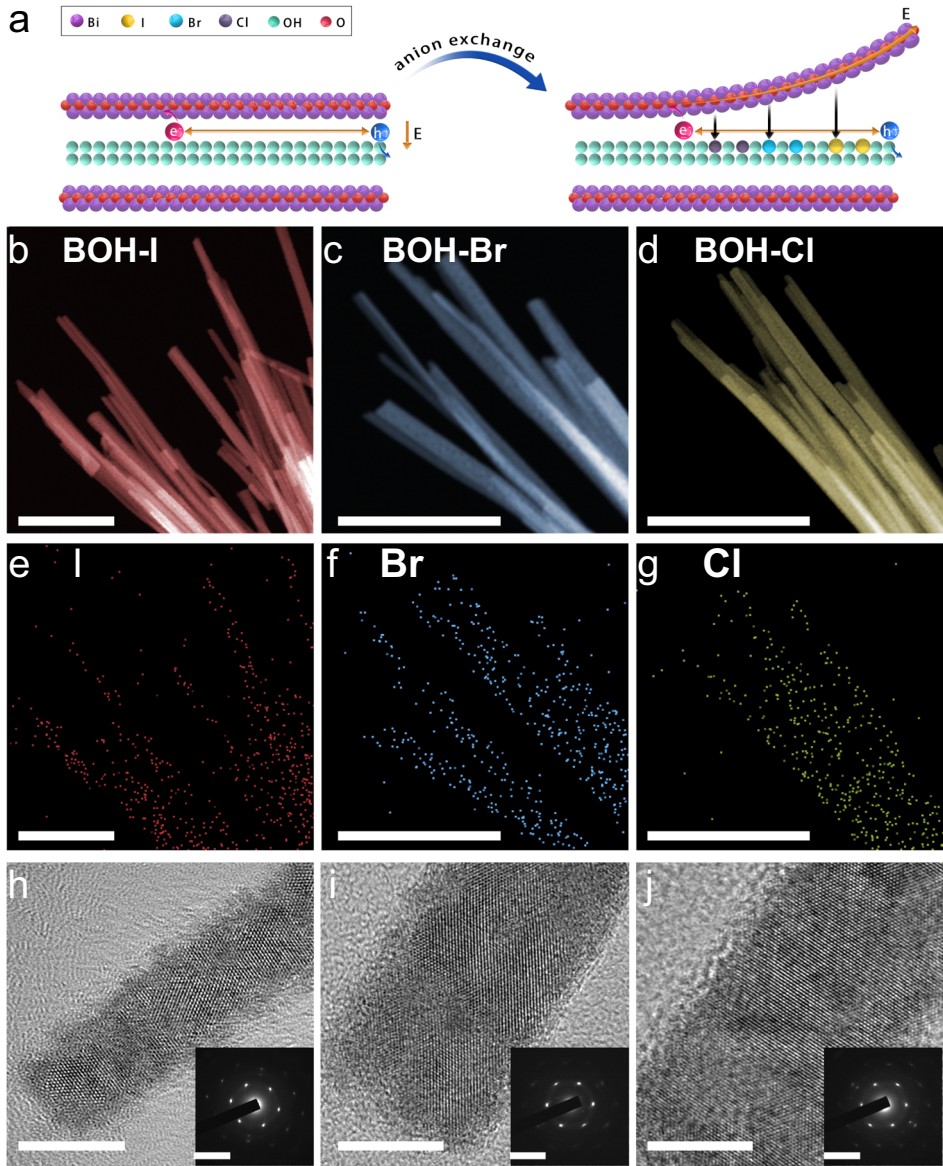

**Fig. 3 Structural characterizations of the BOH-X products obtained via anion exchange. a** Schematic illustration of the halide-anion exchange process. **b–d** Dark-field HRTEM images (scale bar, 200 nm). **e–g** EDS mapping of halogen atoms (scale bar, 200 nm). **h–j** HRTEM-STEM images (scale bar, 10 nm), insets: SAED patterns (scale bar, 0.2 nm$^{-1}$).

the internal electric field intensity of BOH, BOH-Cl, BOH-Br, and BOH-I gradually increased (Fig. 4d). The IEF of BOH-I is double BOH's, while these of BOH-Br and BOH-Cl are 1.6 times and 1.4 times as high as that of BOH, respectively.

**Photocatalytic performances and mechanism.** Imine derivatives are of major importance for the industries of fine chemicals and pharmaceuticals[38–40]. We selected the reaction of visible-light-driven photocatalytic oxidative coupling of benzylamine (Fig. 5a) as the model reaction to assess the effect on catalytic performances induced by halide exchange. Figure 5b shows that with a low ratio (~10%, as mentioned above) of OH$^-$ in pristine BOH replaced by Cl$^-$, Br$^-$, and I$^-$, the conversion of benzylamine is elevated from 44.1% up to 71.0%, 78.3%, and 88.3%, respectively, with the selectivity of 96.3–99.0%, which is consistent with the internal electric field intensity of BOH and BOH-X (BOH < BOH-Cl < BOH-Br < BOH-I). The samples of BOH, BOH-Br, and BOH-Cl have almost identical absorption edges, yet their catalytic performances are rather different, indicating that in this

case the band structure is not a key influencing factor for the catalytic performance. To study the effect of the redshift of the absorption edge for BOH-I on the catalytic performance, four single wavelength lights (365, 405, 450, and 500 nm) were employed for photocatalytic reaction (Supplementary Fig. 27), based on the redshift region of 340–400 nm for BOH-I. The results show that the light absorptions of photocatalyst are almost the same at different wavelengths, the conversion of benzylamine follow the same order (BOH < BOH-Cl < BOH-Br < BOH-I). Hence, it can be concluded that the redshift of BOH-I is not the main factor of its high performance. In addition, products with the increased amount of halogen also show the same photo-catalytic performance trends. The catalytic performance is better with the more halogen introduction (Supplementary Figs. 28–29), indicating that the introduction of halogen ions is indeed the most crucial factor for the improvement of catalytic performance. In addition, we also tested the BOH-nPVP nanosheets (that is, the hydrothermal product obtained without PVP) under identical catalytic conditions (Supplementary Fig. 30), and the sample gave

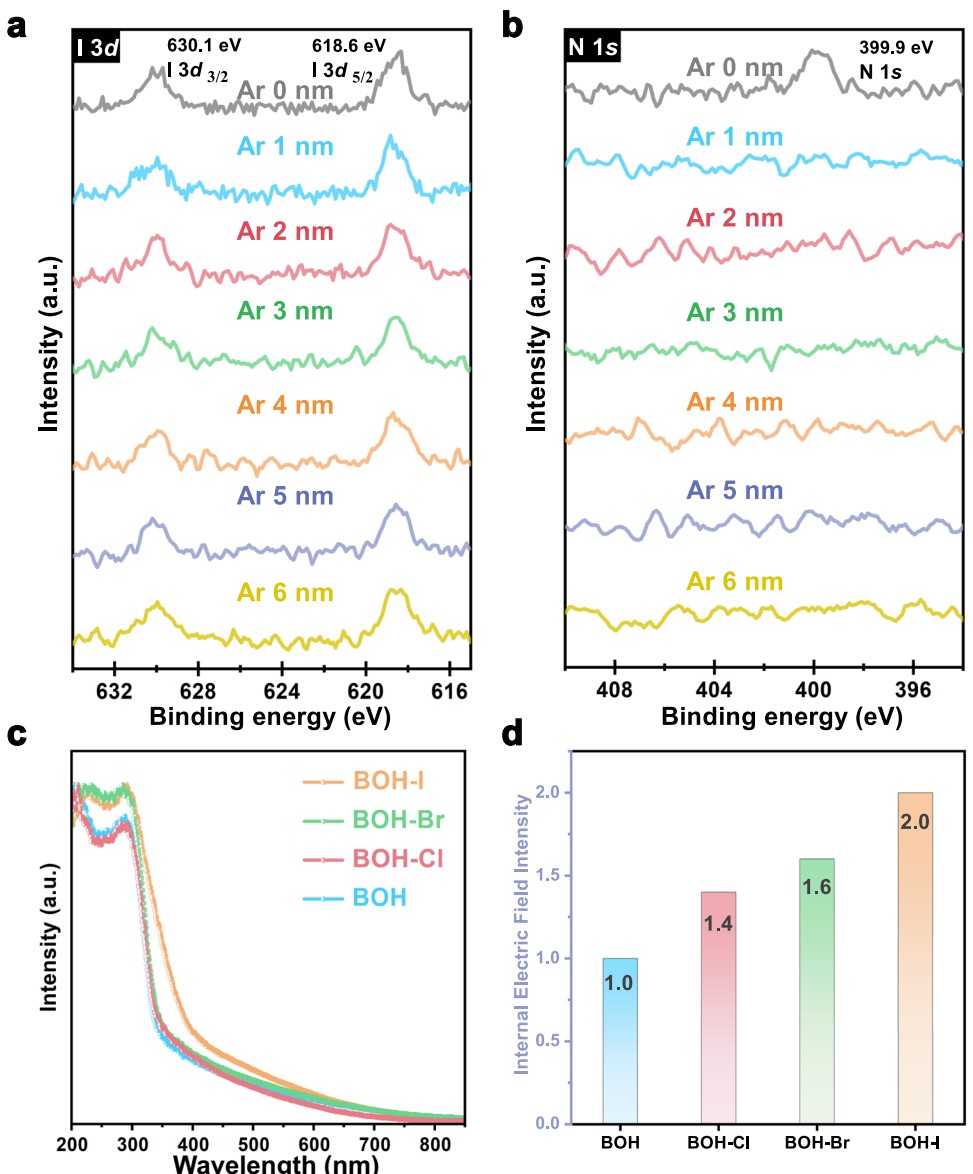

**Fig. 4 Bulk structure, optical property, and internal electric field analysis.** High-resolution XPS spectra with Ar+ sputtering at different depths of **a** I 3*d* and **b** N 1s for BOH-I. **c** UV–vis diffuse reflectance spectra for BOH and BOH-X. **d** Internal electric field Intensity (assuming the intensity of BOH to be "1") for BOH and BOH-X.

a benzylamine conversion of only 6.5%, far lower than that of BOH. This result indicates that the NRs have a superior photocatalytic activity than the nanosheets, probably because the high aspect ratio of the NRs is conducive to the migration and separation of charge carriers. Subsequently, we assessed the durability of the champion sample BOH-I; after five cycles, the benzylamine conversion was well retained at 77.9%, with the selectivity of 98.8% (Fig. 5c). The morphology and microstructure of the catalyst recycled after five runs were also well preserved (as shown in Supplementary Fig. 31). Moreover, BOH-I was used to oxidate other imines (Supplementary Fig. 32). The experiments show that BOH-I have a good catalytic effect on amine substrates (conversion ≥85%, selectivity ≥93%), proving that BOH-I is an excellent catalyst for imine oxidation.

We carried out a series of comparative experiments as well as quenching experiments on the possible active species (Fig. 5d). For example, in the case of BOH-I, the conversion of benzylamine is rather low in the dark or without the photocatalyst, indicating that both the catalyst and light are essential for this reaction.

The conversion in Ar atmosphere was also only marginal, manifesting the essentialness of oxygen. The semiconductor photocatalysts utilize photogenerated electrons and holes to participate in the reaction; the holes can directly oxidize the substrate molecules, and the electrons may reduce molecular oxygen into superoxide radical ($\cdot O_2^-$) to oxidize the substrate. To unveil the reaction mechanism, we added $K_2S_2O_8$ or triethanolamine (TEOA) as the scavenging agent for electrons or holes, respectively. The results show that with either agent introduced, the catalytic performances over all four samples declined. Compared with $K_2S_2O_8$, TEOA would lead to a much more pronounced decline in conversion, particularly for the halide-modified catalysts. However, the effect of $O_2^-$ (often derived from the reduction of oxygen by electrons in organic systems) is not significant according to the EPR tests (Supplementary Fig. 35) and SOD (superoxide dismutase) added experiments (Supplementary Fig. 36). These results suggest that both the electrons and holes function as the active species to participate in the catalytic conversion, the latter playing a major role (See SI for detailed

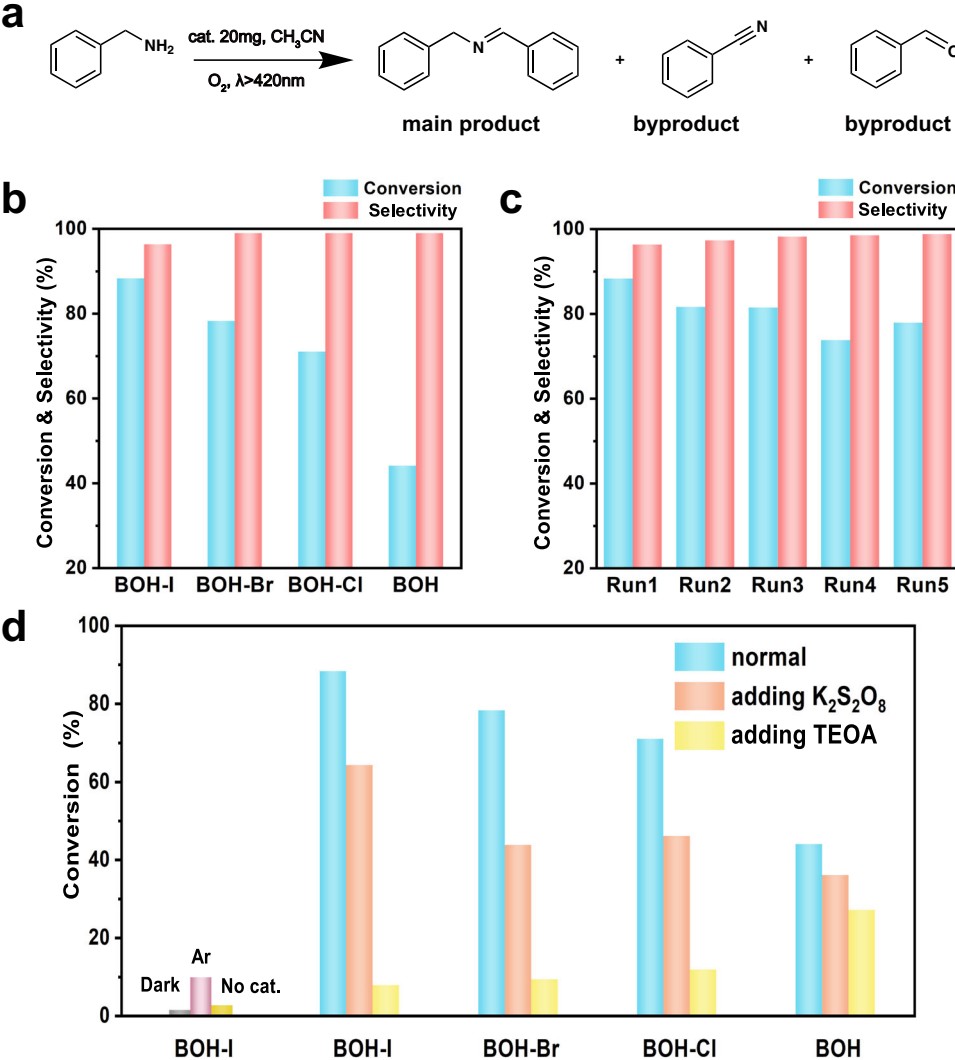

**Fig. 5 Photocatalytic activity. a** Equation of photocatalytic oxidative coupling of benzylamine. **b** Catalytic performances over BOH and BOH-X samples. **c** Results from recycle experiment on BOH-I. **d** Catalytic performances over BOH and BOH-X under normal conditions and with different scavenging agent, and the catalytic performances over BOH-I in dark or under Ar atmosphere.

discussion); moreover, after the halide-anion exchange, the role of holes (which were mainly collected in anionic layers) becomes even more pronounced. To sum up, it can be concluded that the promotion in catalytic performances after halide exchange is due to the enhanced IEF and in turn the elevated efficiency of carrier separation and utilization.

**Structure-activity relationship of BOH and BOH-X.** BOH and three bulk BOH-X models (The calculation model is established with the exchange amount of halogen ions as 20%) were adopted (Fig. 6a–d), with the (001), (100), and (010) facets highlighted for surface cleavage. Owing to the different atomic radii of halides (0.97 Å for Cl, 1.12 Å for Br, 1.32 Å for I), the interlayer spacings of BOH-X are altered (13.6 Å for BOH-Cl, 13.7 Å for BOH-Br, and 13.8 Å for BOH-I), which is in good consistency with the XRD results above.

Furthermore, the calculated DOS (Density of State) confirmed that halide anions in the (001) facet would induce an altered the local electronic structure. Compared with $Cl^-$ and $Br^-$, the introduction of $I^-$ would greatly promote the uneven charge distribution in the cation and anion layers. As shown in the

calculated DOS (Supplementary Figs. 37–40), the $p$ orbital of the introduced halide anion hybridized with the $p$ states of both Bi and O. However, as the energy levels of valence-shell orbitals are different for different halogen atoms, the contributions of these orbitals to the overall band structure are also different (Table 1). Specifically, as the atomic number of the halide anions goes higher, their contribution to the electron density of VBM becomes larger. In contrast, the halide ions have a similar and relatively small contribution (about 0.9%) to CBM, indicating a greater extent of the localization of valence electrons. The result leads to a weakened effect of electronic screening, and is thus beneficial for the separation of electrons and holes, as well as the generation and utilization of holes[2] (which is consistent with the results discussed in the above catalytic mechanism). The localization of valence electrons and the altered interlayer spacing collectively induce a change in the IEF between the cation and anion layers. Our calculations revealed that the electrostatic potential differences in the halide-modified samples are 11.6 eV (for BOH-Cl), 11.9 eV (for BOH-Br) and 12.3 eV (for BOH-I), all higher than the 10.8 eV for pristine BOH (Fig. 6e). This trend is in good accordance with the trend for the catalytic conversion of benzylamine (Fig. 6f). Moreover, the calculated band gaps of

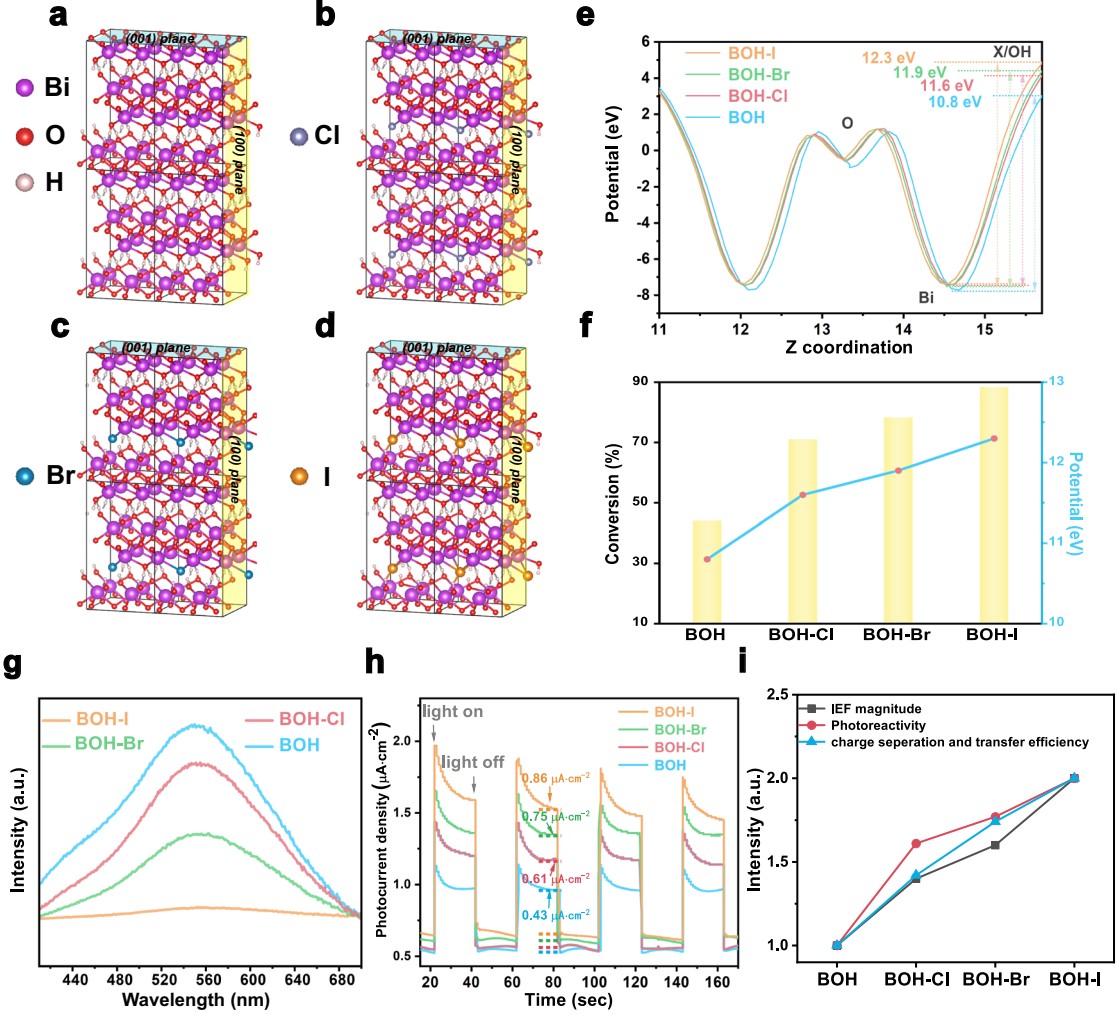

**Fig. 6 Structure-activity relationship of BOH and BOH-X. a–d** Crystallographic structures for BOH, BOH-Cl, BOH-Br, and BOH-I. **e** DFT calculated the local internal electric field for the four catalysts. **f** Superimposed plots of benzylamine conversion (the yellow columns) and calculative IEF intensity (the red dots) for the four catalysts. **g** Steady photoluminescence spectra (excitation wavelength: 346 nm). **h** The photocurrent density spectra (solvent: acetonitrile; bias: 1.25 V). **i** The comparison of charge separation and transfer efficiency, IEF and photoactivity for samples (assuming all the values of BOH to be "1").

**Table 1 The contributions of Bi, O, and X (X = Cl, Br, I) to the near band edges.**

|  |  | Bi (%) | O (%) | X (Cl\Br\I) (%) |
|---|---|---|---|---|
| BOH | VB | 4.2 | 95.8 | – |
|  | CB | 69.2 | 30.8 | – |
| BOH-Cl | VB | 4.5 | 93.5 | 2.0 |
|  | CB | 68.9 | 30.2 | 0.9 |
| BOH-Br | VB | 4.3 | 92.2 | 3.5 |
|  | CB | 68.8 | 30.3 | 0.9 |
| BOH-I | VB | 4.3 | 90.9 | 4.8 |
|  | CB | 68.3 | 30.8 | 0.9 |

BOH and BOH-Cl, BOH-Br, and BOH-I are 3.17 eV, 3.15 eV, 3.13 eV, and 2.98 eV, respectively, showing that the introduction of I− would lead to a more pronounced alteration in the bandgap, in contrast to the cases for Br− and Cl−, which is in good accordance with the UV–vis DRS data.

The enhancement of the IEF intensity endows the photo-catalysts with great potential for efficient charge separation and migration. Then, the carrier separation behaviors of samples were studied. The photoluminescence (PL) spectra (Fig. 6g) of BOH and BOH-X samples show that all four samples give an emission peak at 552 nm. The PL intensities for the BOH-X are all lower than that for BOH, and the intensity decreases with the atomic number of halide anions. The decrease in PL intensity indicates suppression of the recombination of photocarriers. Compared with BOH, halogen-exchanged catalysts all have more robust photocurrent response (Fig. 6h) with the photocurrent density of BOH-I, BOH-Br and BOH-Cl increased successively, which further confirmed the promoting effect of the halogen ions on carrier separation. The increase of IEF intensity of these catalysts is basically consistent with their performance of photocatalytic benzylamine oxidation and the changing trend of photocarriers separation efficiency (Fig. 6i). Therefore, it can be considered that the most crucial reason for the improvement of carrier separation and utilization of BOH-X is the promotion of IEF intensity. Combined with the IEF intensity test experiments, we believe that

as the atomic number of the introduced halide species goes higher, the ionic radius becomes larger, and the charge distribution between the layers become more uneven; the larger electrostatic potential difference between the layers intensifies the interlayer IEF, and thus promotes the carrier separation and utilization.

In conclusion, we report a distinctive Sillenite-structured nanorod material, which features alternating layers of $[Bi_2O_2]^{2+}$ and $OH^-$ ions, and open channels exposed at the surface; on the basis of the pristine nanorods, we have developed an effective strategy to controllably regulate the internal electric field within the material by introducing a low ratio of halide anions to replace the $OH^-$ ions therein. The experimental results and theoretical calculations unveiled the mechanism of IEF regulation: as the atomic number of halide anions goes higher, the spacing between $[Bi_2O_2]^{2+}$ layers exchanges, and the localization of valence electrons becomes more pronounced. This facile method, based on halide-anion exchange with Sillenite-structured compounds, represents a breakthrough from conventional methods on tuning the electronic structures of photocatalysts (such as broadening the absorption range, and modulating band alignment), and circumvents the typical issue (for conventional photocatalysts) of resorting to limited "heterojunction interfaces" to separate photocarriers. Therefore, this method can effectively enhance the efficiency of carrier separation and utilization. We believe that our work here offers new insights into the design and optimization of advanced high-performance photocatalysts.

## Methods

**Materials**. Bismuth nitrate pentahydrate ($Bi(NO_3)_3 \cdot 5H_2O$) was obtained from Aladdin. Potassium chloride (KCl), potassium bromide (KBr), potassium iodide (KI), acetonitrile ($CH_3CN$), triethanolamine (TEOA), potassium peroxydisulfate ($K_2S_2O_8$) and anhydrous ethanol were purchased from Sinopharm Chemical Reagent. Benzylamine was bought from TCI. 2,4-Dichlorobenzenemethanamine and 4-Bromobenzylamine were acquired from 3A, while other amines were obtained from Energy Chemical. Polyvinylpyrrolidone (PVP, M.W. 8000) and mannitol were purchased from Alfa Aesar. Biphenyl was purchased from Acros Organics. N-benzylidenebenzylamine was obtained from Sigma-Aldrich. SOD (Superoxide Dismutase, ≥1400 units/mg dry weight), tetrachloromethane ($CCl_4$), silver chloride (AgCl) and tetramethylammonium hexafluorophosphate were acquired from Aladdin. All chemicals were of analytical grade purity.

**Characterizations**. The crystal phases of BOH and BOH-X (X = I, Br, Cl) were measured using an X-ray diffractometer (XRD, Bruker-D8) with Cu Kα radiation ($\lambda = 1.5418$ Å). Scanning electron microscopy (SEM, Hitachi-SU8010) and transmission electron microscopy (TEM, Hitachi-7700) were used to obtain the structure and morphology of the samples. High-resolution transmission electron microscope (HRTEM, JEM-2010F) of 200 kV was used to acquire more refined structure of the samples. High-angle annular dark-field scanning TEM (HAADF-STEM) was performed on a high-resolution transmission electron microscope (JEM-ARM200F) operated at 300 kV with a probe spherical aberration corrector. The surface properties were characterized by X-ray photoelectron spectroscopy (XPS, ESCALAB-250XI) with a ULVACPHI Quantera microprobe; the bulk properties were analyzed by X-ray photoelectron spectroscopy (XPS, ULVACPHI-Quantera II) with $Ar^+$ spurting. The binding energies were calibrated with respect to the C 1s peak (284.8 eV) from adventitious carbon. UV–visible spectrophotometer (UV-DRS, Hitachi-U3010) was employed to investigate UV–visible diffuse reflectance spectra. The steady photoluminescence spectra were recorded with a fluorescence spectrophotometer (PL, Perkin Elmer-LS55) at room temperature.

**Preparation of BOH nanorods**. The BOH NRs were prepared by a facile one-pot hydrothermal method. Specifically, $Bi(NO_3)_3 \cdot 5H_2O$ (486 mg) was added to deionized water (30 mL), where PVP (400 mg) and mannitol (200 mg) had been added. After stirring for 1 h, the mixture was transferred into a 50 mL Teflon-lined autoclave, and subjected to hydrothermal reaction at 160 °C for 24 h. After cooling, white BOH solid was obtained. The product was washed repeatedly with deionized water and ethanol, and dried in a vacuum oven at 60 °C for 8 h. The theoretical diffraction patterns of BOH were simulated using MAUD (Materials Analysis Using Diffraction).

**Preparation of ion-exchanged samples**. Typically, BOH (60.8 mg) was dispersed in deionized water (4 mL) via sonication, and a solution (4 mL) of KI (0.1 mmol)

was added dropwise. Then, the suspension was sealed into a 12 mL Teflon-lined autoclave, and kept at 60 °C for 12 h. Similarly, the obtained precipitate (BOH-I) was washed repeatedly with deionized water and ethanol, and dried in a vacuum oven at 60 °C for 8 h. BOH-Br, BOH-Cl were prepared in the same way except that KI was replaced respectively with KBr or KCl.

**Photocatalytic reaction**. Typically, the photocatalyst (20 mg) was dispersed in acetonitrile (3 mL) containing amine (0.1 mmol). Then the solution was transferred into a 10 mL quartz tube, with continuous stirring at a proper rotation rate. The photocatalytic reactions were performed with a balloon filled with $O_2$ (~1 atm). After $O_2$ bubbling for 30 min, a 300 W Xenon lamp source (Beijing Perfectlight-Microsolar 300) with a 400 nm cutoff filter was switched on. All the reactions progressed at room temperature with an electric fan. All the filters (including the bandpass filters) were purchased from Beijing Perfectlight. The energy output of the Xenon lamp was approximately 400 mW cm$^{-2}$, which was detected by an optical power and energy meter (Thorlabs- PM100D). After irradiation for 6 h, the mixture was collected and then separated by centrifugation. To identify the reaction product, gas chromatography-mass spectrometry (GC-MS, Thermo Fisher-ISQ system) with an ECD detector (Thermo Trace GC Ultra) was used. Gas chromatography (GC, Thermo Fisher-Trace 1300) with an FID detector was employed to quantitatively analyze the resultant solution, with biphenyl (10 mg) as the internal standard reference.

Active species tests for BOH-I photocatalyst followed the afore-mentioned method, except with extra scavenger agent added. We selected potassium persulfate ($K_2S_2O_8$, 0.1 mmol), triethanolamine (TEOA, 0.1 mmol) and superoxide dismutase (SOD, 1 mg) as scavengers for electrons ($e^-$), holes ($h^+$) and superoxide radicals ($O_2^-$), respectively. In addition, other electronic sacrificial agents such as $CCl_4$, AgCl (0.1 mmol) were also used.

Conversion of benzylamine and selectivity for N-benzylbenzaldimine are defined in the follow equations:

$$\text{Conversion}(\%) = \frac{\sum content\ (mmol)\ of\ each\ product\ analyzed\ via\ GC}{benzylamine\ (mmol)} \times 100\% \quad (1)$$

$$\text{Selectivity}(\%) = \frac{N - \text{benzylbenzaldimine}\ (mmol)}{\sum content\ (mmol)\ of\ each\ product} \times 100\% \quad (2)$$

**EPR test**. To capture the signal of $\cdot O_2^-$ radicals, electron paramagnetic resonance spectra were recorded on an electron paramagnetic resonance spectrometer (EPR, JEOL FA-200). Typically, the catalyst sample (5 mg) and benzylamine (0.2 mmol) were dispersed in acetonitrile (1 mL) with 5, 5-dimethyl-1-pyrroline N-oxide (DMPO, 10 μL). The mixture was transferred into a paramagnetic tube. Upon irradiation with a 300 W Xe lamp ($\lambda > 420$ nm), the ESR spectra were recorded.

**Electrochemical and photoelectrochemical measurements**. The catalyst (5 mg) was dispersed in ethanol (1 mL). The mixture was deposited dropwise on the pretreated indium tin oxide (ITO) wafer and then dried for 24 h at ambient temperature. The Mott–Schottky experiments were conducted to evaluate the band positions of BOH and BOH-X, and were completed within the potential range from −0.7 to −0.1 V at a frequency of 500 Hz. Measurements were performed on an electrochemical workstation (PMC-500, Princeton) with a standard three-electrode system. The catalyst was used as the working electrode, while a Pt foil and Ag/AgCl electrode served as the counter electrode and the reference electrode, respectively. Tetramethylammonium hexafluorophosphate was used as the electrolyte (0.1 mol L$^{-1}$) and acetonitrile was used as the solvent for the photocurrent test. Other tests employed an aqueous $Na_2SO_4$ solution (0.1 mol L$^{-1}$) as the electrolyte.

**Theoretical calculations**. Theoretical calculations were performed using Vienna ab initio simulation packages (VASP) based on density functional theory[41]. Interactions between core and valence electrons were described by the projector augmented wave (PAW) pseudopotentials[42]. The generalized gradient approximation (GGA) in the scheme proposed by Perdew, Burke, and Ernzerhof (PBE) was adopted to express the electron exchange correlation with a cutoff energy of 400 eV, while the van der Waals effect and hydrogen bonding interactions were accounted for by the DFT-D3[43]. All atoms were fully relaxed in z dimensions till all residual forces have declined below 0.02 eV Å$^{-1}$ and the convergence of energy was set to $1 \times 10^{-5}$ eV. The Brillouin zone was sampled by the Monkhorst–Pack method with a 6× 6× 6 k-point grid[44]. To obtain a more appropriate description of the electronic properties of BOH and BOH-X here considered, we also performed reference calculations using the hybrid functional proposed by Heyd, Scuseria, and Ernzerhof (HSE06)[45,46].

## Data availability
The authors declare that all the important data to support the findings in this paper are available within the main text or in the Supplementary information. Extra data are available from the corresponding author upon reasonable request.

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

## Acknowledgements

This work was supported by the National Key R&D Program of China (2017YFA0700101, 2016YFA0202801), the National Natural Science Foundation of China (21971135, 21925202, 21872076, 21590792), Beijing Natural Science Foundation (JQ18007). Also, we acknowledge the support of the Analysis Center of Tsinghua University for XPS measurements. And we thank Dr. C. Zhang for the help in preparing this manuscript.

## Author contributions

T.H., X.C., Q.P. and Y.L. conceived the idea. T.H., X.C. and W.-C.C. carried out the sample synthesis, characterization and electrochemical test. K.S., A.H. and C.C. conducted the DFT calculation. C.Y. performed the XRD simulation of the material. T.H., X.C. and Q.P. wrote the manuscript. R.L., Z.Z., X.T. and D.W. discussed the electro-catalytic measurements and gave useful suggestions. Z.C. and D.Z. discussed the organic reactions and gave useful suggestions. Q.P., C.C. and Y.L. were responsible for the overall direction of the project. All the authors contributed to the overall scientific interpretation and edited the manuscript.
