## [Peer Review File · Nature Communications]

REVIEWER COMMENTS

Reviewer #1 (Remarks to the Author):

Li et al report fabrication of nanorods of layered structure composed of $(\text{Bi}_2\text{O}_2)_2^+$ and charge compensating hydroxyl ions with high cross-section-to-length aspect ratio via solvothermal approach employing polyvinylpyrrolidone and mannitol as directing agents. Using ion exchange reaction, hydroxyl groups have been partially replaced by halide anions (ca 10%). The manuscript is well structured and written. It has been concluded that substitution of hydroxyls with halide ions induces formation of internal electric field in the bulk of the material and as a result leads to improved charge separation and improved performance in photocatalysis. This is the main point that authors appeal in the present work and which may, in principle, justify publication in top chemistry journal such as Nature Communications. However, evidences supporting such claims have been derived mainly from steady state and time resolved emission spectroscopy. Despite materials have been characterized by a set of techniques, some questions related to the structure remain open. Overall, the manuscript offers synthesis, characterization and application of inorganic semiconductor of peculiar morphology in photocatalysis. However, authors should provide more evidences supporting formation of IEF as claimed in this manuscript.

1. To propose a crystal structure, authors characterized materials by TEM, XPS and X-Ray diffraction. They found that BOH does not contain nitrate anions, but nevertheless $\text{Bi}_2\text{O}_2(\text{OH})(\text{NO}_3)$ formula has been assigned to the material. Secondly, X-Ray of BOH does not match to $\text{Bi}_2\text{O}_2(\text{OH})(\text{NO}_3)$ (Figure S4). These points related to the material structure should be clarified. Did authors attempt to simulate diffraction pattern of BOH taking into account experimental data acquired using different techniques to propose more reasonable structure?
2. In this paper authors synthesized materials with ca. 10% of halide through ion exchange. What effect would substitution have if more halide anions is introduced into the material? Dependency of material property versus composition would strengthen this work.
3. Line 213. Halide anions can be hardly called highly electronegative. Electronegativity of all halides is lower compared to oxygen, while for iodide it is comparable to carbon.
4. Line 238-240. I would expect that longer lifetime of photogenerated charge carriers is beneficial for photocatalysis.
5. Line 253. It is not clear how CBM values from flat-band potential have been obtained.
6. Line 261. In this manuscript authors study synthesis of imines rather than imides.
7. Potassium persulfate has been used as electron acceptor. However, oxidation power of this reagent is sufficient to enable oxidation of benzylamine without any semiconductor and light irradiation.[European Journal of Organic Chemistry, 2019(6), 1242-1250; Bulletin of the Chemical Society of Japan, 2019, 63(1), 301-3]
8. Line 281-282. It is inferred that rod-like morphology of BOH is beneficial for improved performance in photocatalysis. But how about surface area of these materials?
9. From the description of the model, used for theoretical calculations, it is not clear if BOH, in which all OH have been substituted with the respective halide anions or only a fraction of them, was used.
10. DOS. Does zero correspond to Fermi level?
11. Line 328-329, 'the p orbital of the introduced halide anion hybridized with the p states of both Bi and O.' It is not obvious neither from Fig 6e nor S19-22. Perhaps partial density of states for elements would

be more illustrative.

12. Line 329-330, 'The introduction of I⁻ would lead to a more pronounced alteration in the bandgap,' Does this sentence refer to Table 1?

13. Figure 6. Explicit numbers corresponding to the colors of the legend would be beneficial.

14. In the introduction, it is emphasized that low efficiency of carriers separation hampers development of sophisticated organic reactions. It was expected that in this work, using the developed semiconductors, authors would give an example of 'sophisticated organic reaction'. However, application of the material is limited to oxidation of benzylamine only, which is rather a trivial reaction.

Once the abovementioned comments are positively addressed, the manuscript may be considered for publication in Nature Communications.

Reviewer #2 (Remarks to the Author):

The manuscript entitled "Anion-Exchanged-Mediated Internal Electric Field: Boosting Photogenerated Carrier Separation and Utilization" presents the synthesis of bismuthoxy-hydroxyde as nanorods and the effect of halogen substitution of the hydroxides on its photocatalytic abilities. The authors claim that the partial substitution of OH⁻ by halogen atoms enhance the photocatalytic conversion due to the formation of a strong internal electric field. This is an interesting work, but the results and hypothesis are too preliminary for publication in Nature Communication.

Here is the list of my remarks:

- 1- One main reason of the formation of an IEF presented by the author is the decrease of the photoluminescence efficiency. But a decrease of photoluminescence efficiency in solids is more attributed to trap states in the bandgap than an efficient charge carrier dissociation. Thus, this argument cannot be used as it is to prove the existence of an IEF.
- 2- Another argument given to support the idea of an IEF comes from the computations. But these computations are not at the state of the art. The author should look at the literature about the most appropriate computational protocol to characterize a semiconductor for photocatalytic application. They should use the range separated hybrid functional HSE06 for their calculation and deeply characterize their materials (bandgap, dielectric constant, effective masses). Furthermore, the Figure 6 is absolutely unclear. What information do we extract from the charge density? Figure 6f is too small to see the differences. From the Table 1, it appears that the DOS is almost unchanged upon halogen substitution. How do the authors explain the apparition of an IEF? Clearly, the DFT part could be much stronger and, if placed earlier in the paper, could support more the conclusion.
- 3- Figure 4d, the authors should give the band positions with respect to an electrode reference.
- 4- The authors should provide the response spectra as a function of the wavelength to see if the increase of efficiency of the BOH-I is not only due to a redshift of the absorption spectra.
- 5- On figure 5a, the nitrile group on the side product is linear. Please correct the structure and the TOC that also contains this structure.

Point-by-point response

Replies to the Reviewer #1

Li et al. report fabrication of nanorods of layered structure composed of $(\text{Bi}_2\text{O}_3)^{2+}$ and charge compensating hydroxyl ions with high cross-section-to-length aspect ratio via solvothermal approach employing polyvinylpyrrolidone and mannitol as directing agents. Using ion exchange reaction, hydroxyl groups have been partially replaced by halide anions (ca 10%). The manuscript is well structured and written. It has been concluded that substitution of hydroxyls with halide ions induces formation of the internal electric field in the bulk of the material and, as a result leads to improved charge separation and improved performance in photocatalysis. This is the main point that authors appeal in the present work and which may, in principle, justify publication in top chemistry journal such as Nature Communications. However, evidences supporting such claims have been derived mainly from steady state and time resolved emission spectroscopy. Despite materials have been characterized by a set of techniques, some questions related to the structure remain open. Overall, the manuscript offers synthesis, characterization and application of inorganic semiconductor of peculiar morphology in photocatalysis. However, authors should provide more evidences supporting formation of IEF as claimed in this manuscript.

Reply: We really appreciate your positive feedback and constructive suggestions for our manuscript. Following your advices and suggestions, we have revised the manuscript carefully.

The lack of solid IEF evidence is indeed the deficiency of the original manuscript. Therefore, we conducted supplementary experiments to explore the internal electric field intensity for BOH and BOH-X, hoping to provide direct evidence for the existence of IEF. The method of measuring IEF intensity reported by Kanata et al. has been widely adopted, according to Photoreflectance signal amplitude^{R1-R6}. It was based on the effect of internal electric field disturbance on the dielectric constant of the semiconductor so that the photoreflectance signal was generated.

The formula is as follows:

$$F_s = (-2V_s\rho/\epsilon\epsilon_0)^{1/2}$$

Where F_s is the internal electric field magnitude, V_s is the surface voltage, ρ is the surface charge density, ϵ is the low-frequency dielectric constant, and ϵ_0 is the permittivity of free space. As ϵ and ϵ_0 are constants, the IEF intensity is mainly determined by the surface voltage (V_s) and the surface charge density (ρ). Therefore, we could qualitatively compare the internal electric field intensity of BOH and BOH-X according to the $(V_s\rho)^{1/2}$ values.

Based on that, we first applied the open-circuit potentials measurements to evaluate the surface voltages of BOH and BOH-X. As shown in Figure R1, the surface voltage of BOH-I is 0.268 V, which is greater than that of BOH-Br (0.198 V), BOH-Cl (0.175 V), and BOH (0.142 V), indicating the enhanced IEF.

Figure R1. The open-circuit potentials of BOH (a) and BOH-X (b–d).

As Le Formal and Gratzel et al. reported, the accumulated positive charge on the surface is proportional to the integral value, calculated from the transient photocurrent density minus the steady-state photocurrent density in the same time^{R7}. Therefore, the transient photocurrent density measurements were conducted, controlling the same contact areas with ITO of every sample. The surface charge densities of BOH and BOH-X were then obtained by the integral of the transient anodic photocurrent peaks (Figure R2). As expected, the integral value of the photocurrent response of BOH-I ($148.1 \mu\text{c}\cdot\text{cm}^{-2}$) is the maximum among all specimens, which is even twice as magnitude as that of BOH ($70.4 \mu\text{c}\cdot\text{cm}^{-2}$), and also higher than that of BOH-Br ($123.8 \mu\text{c}\cdot\text{cm}^{-2}$) and BOH-Cl ($105.4 \mu\text{c}\cdot\text{cm}^{-2}$).

It can be found that the internal electric field intensity of BOH, BOH-Cl, BOH-Br, and BOH-I gradually increased (Figure R3). Taking the IEF intensity of BOH as the “1”, the IEF intensity of BOH-I is double BOH’s, while these of BOH-Br and BOH-Cl are “1.6” and “1.4” respectively. Moreover, the increase of the internal electric field intensity is basically consistent with the photocatalytic performance of these catalysts for benzylamine oxidation (that is, the photocatalytic activity of BOH is “1”, the catalytic activities of BOH-Cl, BOH-Br, and BOH-I are “1.6”, “1.8” and “2.0” respectively). It further proves the validity of our strategy to regulate the IEF intensity through the exchange of halogen ions and directly demonstrates the positive correlation between the IEF intensity and the photocatalytic performance.

The above-related content has been supplemented in “**Results: Synthesis strategy and characterization of BOX nanorods (X = Cl, Br, I).**” and the corresponding SI section.

Figure R2. The transient photocurrent density of BOH (a) and BOH-X (b–d).

Figure R3. The internal electric field intensity and catalytic performance of BOH and BOH-X (assuming both the IEF intensity and the catalytic activity of BOH to be “1”).

Comment 1: To propose a crystal structure, authors characterized materials by TEM, XPS and X-Ray diffraction. They found that BOH does not contain nitrate anions, but nevertheless $\text{Bi}_2\text{O}_2(\text{OH})(\text{NO}_3)$ formula has been assigned to the material. Secondly, X-Ray of BOH does not match to $\text{Bi}_2\text{O}_2(\text{OH})(\text{NO}_3)$ (Figure S4). These points related to the material structure should be clarified. Did authors attempt to simulate diffraction pattern of BOH taking into account experimental data acquired using different techniques to propose more reasonable structure?

Reply: Thanks for pointing it out. We are very sorry that there may be some improper expressions in our previous manuscript, leading to misinterpretation. In fact, we did not identify BOH as $\text{Bi}_2\text{O}_2(\text{OH})(\text{NO}_3)$. The latter was mentioned only because it has a similar $[\text{Bi}_2\text{O}_2]^{2+}$ cation skeleton structure to BOH (according to the similar adjacent Bi-Bi distances of BOH and the known compound $\text{Bi}_2\text{O}_2(\text{OH})(\text{NO}_3)$), which does not mean that BOH has the chemical formula of $\text{Bi}_2\text{O}_2(\text{OH})(\text{NO}_3)$. The different types of anions between BOH and $\text{Bi}_2\text{O}_2(\text{OH})(\text{NO}_3)$ make their cells' structure much different. Specifically, NO_3^- anion has a larger radius (2.00 Å) than OH^- (0.89 Å), so when OH^- anions are intercalated between the $[\text{Bi}_2\text{O}_2]^{2+}$ layers (in the case of BOH) instead of NO_3^- (in the case of $\text{Bi}_2\text{O}_2(\text{OH})(\text{NO}_3)$), the bridging effect of the anions would become less pronounced, leading to increased spacing between neighboring $[\text{Bi}_2\text{O}_2]^{2+}$ layers owing to coulombic repulsion^{R8}. And the altered symmetry for BOH also results in diffraction peaks located at different angles from those for $\text{Bi}_2\text{O}_2(\text{OH})(\text{NO}_3)$, as well as more diffraction peaks.

This new structure of BOH has never been reported, so we did not find a suitable standard XRD pattern corresponding to it. According to the reviewer's suggestion, we fitted the XRD pattern of $\text{Bi}_2\text{O}_2(\text{OH})_2$, as shown in Figure R4. It is observed that the BOH is basically in line with the fitted figure, and the prominent diffraction peaks (marked as peak 1-5) are consistent one-to-one match. Furthermore, a weaker diffraction peak shoulder near peak 1 might arise from the lattice distortion^{R9}. In addition, the diffraction peaks * do not exist in the simulated results. However, it disappeared after drying (dissolving BOH powder in a small amount of

ethanol and redrying the sample at 100°C for 15h) (Figure R5), so we speculate the diffraction peak * might come from the hydrate impurity. However, the catalytic experiment shows that impurity has no noticeable effect on the catalytic activity (Figure R6). These results indicate that the structure we inferred ($[\text{Bi}_2\text{O}_2]^{2+}$ is connected by OH^- between layers) is reasonable. Combined with the XPS experiments and the TEM data, we further clarify that the basic composition of the BOH structure is $\text{Bi}_2\text{O}_2(\text{OH})_2$.

The above-related content has been supplemented in “**Results: Synthesis strategy and characterization of BOH nanorods.**” and the corresponding SI section.

Figure R4. The XRD patterns of the BOH and simulated $\text{Bi}_2\text{O}_2(\text{OH})_2$, with the assumed crystal cell ($a=6.74 \text{ \AA}$; $b=5.8 \text{ \AA}$; $c=22.2 \text{ \AA}$; $\alpha=90^\circ$; $\beta=90^\circ$; $\gamma=90^\circ$).

Figure R5. The XRD patterns of the BOH (before and after drying treatment) and simulated Bi₂O₂(OH)₂.

Figure R6. Catalytic performances of BOH (before and after drying treatment).

Comment 2: In this paper authors synthesized materials with ca. 10% of halide through ion

exchange. What effect would substitution have if more halide anions is introduced into the material? Dependency of material property versus composition would strengthen this work.

Reply: Thanks for your valuable suggestion. In the original manuscript, we introduced 0.1 mmol potassium halide, and XPS semi-quantitatively showed that exchange ratio was about 10%. We hydrothermally synthesized ion-exchanged products with more (0.2 mmol-2.0 mmol) potassium halide in the supplementary experiments. The TEM (Figure R7-9) characterization shows little differences in the morphology of these ion-exchanged products. It is worth noting that when 20 times (2 mmol) of X^- ions introduced, the XRD patterns (Figure R10) does not change significantly compared with the original BOH-X, indicating the limited exchange capacity of halide ions which is further proved by XPS semi-quantitative analysis afterwards (Figure R11). Specifically, when the amount of potassium halide increases from 0.1 mmol to 2 mmol (reached to 20 times), the content of halogen actually exchanged into BOH only increases from 10% to 17%. As the amount of potassium halide increases, the amount of halogen ions actually introduced increases gradually slowly.

Figure R7. The TEM characterization of ion-exchange products with different amounts of Cl ions.

Figure R8. The TEM characterization of ion-exchange products with different amounts of Br ions.

Figure R9. The TEM characterization of ion-exchange products with different amounts of I ions.

Figure R10. The XRD characterization of ion-exchange products with increased amounts of Cl^- (a), Br^- (b), I^- (c).

Figure R11. The relationship between the amount of KI added in the synthesis and the actual introduction of I⁻ (from the XPS semi-quantitative experiments).

The benzylamine oxidation experiments (Figure R12-13) were conducted with these catalysts. It can be found that for the same halogen ion, the greater the amount of halogen ions introduced, the better the catalytic performance; For Cl, Br, and I ions, the trend of original catalytic performance can still be maintained (Cl < Br < I) with more halogen ion introduced, which further supports the central thesis of our work that halogen ion-exchange could facilitate the separation and utilization of photocarriers.

The above-related content has been supplemented in “**Results: Synthesis strategy and characterization of BOX nanorods (X = Cl, Br, I). & Photocatalytic performances and mechanism**” and the corresponding SI section.

Figure R12. The photocatalytic performances of catalysts with more halogen ions.

Figure R13. The introduction of I⁻ and the photocatalytic performance plotted versus the amount of KI added in the synthesis.

Comment 3: Line 213. Halide anions can be hardly called highly electronegative. Electronegativity of all halides is lower compared to oxygen, while for iodide it is comparable

to carbon.

Reply: Thanks for pointing it out. The expression of “electronegativity” is indeed not appropriate, and we have deleted the relevant expression in the manuscript.

Comment 4: Line 238-240. I would expect that longer lifetime of photogenerated charge carriers is beneficial for photocatalysis.

Reply: Thanks for pointing it out. The actual separation and migration process of carriers is much complicated. Some researchers believe that the shorter the transient fluorescence lifetime of the material, the stronger the carrier separation capability^{R10-R13}. In addition, the lifetime comparisons of the transient fluorescence spectra of solid samples might not adequately account for the carrier separation mechanism in the actual real photocatalytic situation (that is, the solvent effect cannot be ignored.). At this point, we designed new experiments. Photoelectrochemical measurements were carried out on BOH and BOH-X with acetonitrile solvent (Figure R14), and carriers separation efficiency could be evaluated by comparing the photoelectric response. The photocurrent density of BOH-I, BOH-Br, BOH-Cl, and BOH decreases successively, indicating the gradually decreasing separation efficiency of carriers.

The above-related content has been supplemented in “**Results: Structure-activity relationship of BOH and BOH-X.**” and the corresponding SI section.

Figure R14. The photocurrent density of BOH and BOH-X. (Acetonitrile as the solution, tetramethyl hexafluorophosphate amine as the electrolyte, bias: 1.25 eV.)

Comment 5: Line 253. It is not clear how CBM values from flat-band potential have been obtained.

Reply: Thanks for pointing it out. In the previous manuscript, several studies suggest that the CBM values for n-type semiconductors can be approximated as the Fermi level^{R14}, and thus the band structures can be estimated. However, based on your comment, we have redesigned the experiment regarding some research to obtain more accurate BOH and BOH-X band structures. The specific methods are as follows:

First, we measured the XPS valence band spectrum of each catalyst (Figure R15(a–c)) and obtained the distances from the VBM of materials to the Fermi level as 1.92 eV, 1.90 eV, 1.88 eV, and 1.62 eV corresponding to BOH, BOH-Cl, BOH-Br, and BOH-I respectively. According to the Tauc plot (Figure R15(d)) obtained by UV-vis DRS conversion, the band gaps of BOH, BOH-Cl, BOH-Br, and BOH-I are 3.60 eV, 3.56 eV, 3.55 eV, and 3.13 eV separately. Correspondingly, the distances from the CBM to the Fermi level are calculated to

be 1.68 eV, 1.66 eV, 1.67 eV, and 1.51 eV for BOH, BOH-Cl, BOH-Br, and BOH-I, respectively (using the equation $E_{CB} = E_g - E_{VB}$).

In order to assess the Fermi levels for BOH and BOH-X, we conducted the Mott-Schottky measurements at the potential of -0.1 V to -0.7 V (at a fixed frequency of 500 Hz). Based on the Mott-Schottky formula, $C_{sc}^{-1} = 2(\Delta\Phi_{sc} - RT/F) (q\epsilon\epsilon_0N)^{-1}$ (where $\Delta\Phi_{sc} = V - V_{fb}$, V_{fb} is the flat band potential, T is the Kelvin temperature, F is the Faraday constant, R is the gas constant, ϵ and ϵ_0 are the semiconductor dielectric constant and vacuum dielectric constant, respectively, q is the charge quantity, and N is the doping concentration), we plotted C_{sc}^{-1} versus V (Figure S22), and then obtained the flat bands of BOH, BOH-Cl, BOH-Br and BOH-I through the intercept of the abscissa $V_0 = V_{fb} + RT/F$. Therefore, the potentials of BOH, BOH-Cl, BOH-Br and BOH-I are 0.15 eV, 0.21 eV, 0.03 eV and 0.02 eV (versus Ag/AgCl at pH 6.80) respectively, which are -0.25 eV, -0.19 eV, -0.17 eV and -0.38 eV separately, relative to the normal hydrogen electrode at pH=0 (NHE). Since the Fermi level E_f and V_{fb} have the same value, combined with the valence band spectrum and UV-vis DRS results, the energy band diagrams of BOH and BOH-X relative to the normal hydrogen electrode are finally obtained as Figure R16.

The above-related content has been supplemented in “**Results: Structure-activity relationship of BOH and BOH-X.**” and the corresponding SI section.

Figure R15. The characterization of BOH and BOH-X: (a–c) valence band spectra; (d) Tauc plots.

Figure R16. The schematic diagram of band structure of BOH and BOH-X.

Comment 6: Line 261. In this manuscript authors study synthesis of imines rather than imides.

Reply: We thank the reviewer for the careful examination of our paper. In the Page X, Line X, "Imides" was corrected to "imines". Besides, we have re-examined our manuscript and the supporting information and made some revisions to the wording and grammar. The revised parts are listed below.

Corresponding changes:

1. Page 2, Line 7, and Page 4, Line 9: " basic bismuth nitrate compound" was corrected to " bismuth oxyhydroxide compound".
2. Page 6, Line 10: "NRs" was corrected to "nanorods".
3. Page 6, Line 17: "with" was added.
4. Page 11, Line 17: "1 eV" was corrected to "0.2 eV".
5. Page 15, Line 7: "Imide" was corrected to "Imine".
6. Page 26: the by-products, including benzaldehyde and benzonitrile, were added to TOC (Scheme 1).
7. Page 34, Line 3: "catalytic" was corrected to "Catalytic".
8. Page 34, Line 4: " results of" was corrected to " Results from".
9. Page 34, Line 4: "catalytic" was corrected to "Catalytic".
10. Page 34: the structural formula of benzonitrile in Figure 5(a) was corrected.
11. Page 40, Line 17: " stirred" was corrected to " stirring".
12. Page 43, Line 20: " of pro-posed" was corrected to " proposed ".

Comment 7: Potassium persulfate has been used as electron acceptor. However, oxidation power of this reagent is sufficient to enable oxidation of benzylamine without any semiconductor and light irradiation. [European Journal of Organic Chemistry, 2019(6), 1242-1250; Bulletin of the Chemical Society of Japan, 2019, 63(1), 301-3]

Reply: Thanks for your recommendation for these two references. In the supplementary, we added some typical electron sacrificial agents to the reaction system with only substrate and solvent (without photocatalyst) and repeated the benzylamine oxidation experiment (Figure R17 left). It is found that these electron scavengers themselves could promote the conversion of benzylamine to a certain extent, and the conversion of benzylamine with $K_2S_2O_8$ used in the manuscript is 23.8%.

Then we selected CCl_4 , which has the worst catalytic performance of benzylamine oxidation with the conversion of 8.5%, as the electron quenching agent to explore the role of electrons in photocatalysis (Figure R17, right). For BOH, the oxidation of benzylamine is promoted with CCl_4 added, while for BOH-X, the reactions are inhibited to different degrees. Nevertheless, inhibition is weaker than that with $K_2S_2O_8$ added. In combination with the catalytic performance of these two electron scavengers, it can be considered that electrons also participate in benzylamine oxidation for ion-exchanged catalysts. However, it is indeed not rigorous to directly conclude that “the role of holes is more outstanding than that of electrons in the catalytic process of BOH and BOH-X, and the effect of holes is stronger for BOH-X than that of BOH” only from previous experiments with sacrificial agents. ”.

Figure R17. Left: The photocatalytic performance of different electronic sacrificial agents on the oxidation of benzylamine (without additional catalyst); Right: The photocatalytic performance over BOH and BOH-X with CCl₄ and K₂S₂O₈ as electron capturer.

However, the EPR and corresponding catalytic experiments in the original manuscript could mediate verify that "electrons do not play a dominant role in the photocatalytic process". It is generally believed that for the photo-oxidation with oxygen participated in organic solvents, the utilization of electrons lies in the fact that the superoxide radicals, which are obtained by electron reduction of oxygen, have excellent oxidizing ability^{R15}. The EPR experiments on superoxide radical show that all the samples can generate superoxide free radicals (Figure R18 left), but BOH-X produce more superoxide free radicals. According to the comparative experiment of superoxide radicals quenching (right of Figure R18), after superoxide dismutase (SOD) was added, the benzylamine conversion didn't decline significantly (74.1% with SOD added). It proves that the oxidation by superoxide radicals is not the main pathway. Therefore, we have concluded that direct hole oxidation is dominant to a certain extent, which is consistent with the original manuscript.

The above-related content has been supplemented in **“Results: Photocatalytic performances and mechanism.”** and the corresponding SI section.

Figure R18. Left: EPR spectra for the samples before and after ion exchange, with DMPO as scavenger. Right: Catalytic performances of the normal BOH-I and after addition of superoxide dismutase (SOD).

Comment 8: Line 281-282. It is inferred that rod-like morphology of BOH is beneficial for improved performance in photocatalysis. But how about surface area of these materials?

Reply: We thank the reviewer for the valuable suggestion. In order to explore the influence of specific surface areas of BOH and BOH-X in the process of photocatalysis, we performed the BET tests (Figure R19). The specific surface area of BOH nanorods increased from 11.7 m²/g to 31.7 m²/g compared with the hydrothermal sheet product without PVP (BOH-NPVP). However, the specific surface areas of the catalyst before and after ion exchange are the same (BOH: 31.7 m²/g; BOH-Cl: 30.9 m²/g; BOH-Br: 29.4 m²/g; BOH-I: 30.1 m²/g), so the influence of specific surface area change on BOH-X-ray catalytic performance can be excluded.

The above-related content has been supplemented in “**Results: Synthesis strategy and characterization of BOH nanorods. & Synthesis strategy and characterization of BOX nanorods (X = Cl, Br, I).**” and the corresponding SI section.

Figure R19. The specific surface areas of BOH-nPVP and the samples before and after ion exchange.

Comment 9: From the description of the model, used for theoretical calculations, it is not clear if BOH, in which all OH have been substituted with the respective halide anions or only a fraction of them, was used.

Reply: Thank you for pointing this out. We are sorry for missing such important information in the previous manuscript. In order to enable the theoretical calculation results more in line with the actual situation, as well as to make the intrinsic differences of halogen ion exchange more intuitive, 20% halogen ion exchange was chosen for the theoretical study (while the halogen ion exchange capacity measured in our experiment is about 10%).

To investigate the influence of halogen ions more professionally, we adopted the suggestion of reviewer #2 and recalculated with the HSE06 function, which is more suitable for photocatalysis. The amount of halogen ion exchange is still 20%. The proportion of halogen ions in the calculation model has been supplemented in the calculation part of the manuscript.

According to calculation using the HSE06 function, the hybridization of p orbital of halogen atom with p orbital of Bi and O is more pronounced, contributing halogen atoms to the valence band more prominent, which more powerfully proves that the introduction of halogen could further strengthen the IEF intensity of the materials.

Comment 10: DOS. Does zero correspond to Fermi level?

Reply: Yes, the Zero indeed corresponded to the Fermi Level. The DOS calculations using new HSE06 function were further taken, and the zero also corresponds to the Fermi level.

Comment 11: Line 328-329, ‘the p orbital of the introduced halide anion hybridized with the p states of both Bi and O.’ It is not obvious neither from Fig 6e nor S19-22. Perhaps the partial density of states for elements would be more illustrative.

Reply: Thanks for your suggestion, in which the partial density of states for elements should be shown. We are sorry that we did not superposition DOS of each element in the original manuscript, making it difficult for readers to see the orbital hybridization between halogen atoms and BOH visually. For this reason, after recalculating with the HSE06 function, we superimposed the DOS of halogen elements over Bi and O, as shown in Figure R20-23. It can be seen that the p orbital of halogen atoms is more obviously hybridized with the p-orbital of Bi and O. We have updated the DOS diagrams in the SI section.

Figure R20. (a) The calculated density of state of BOH; (b) and (c) The calculated partial density of state of bismuth and oxygen, respectively.

Figure R21. (a) The calculated density of state of BOH-Cl; (b) The calculated partial density of state of bismuth, oxygen and chlorine element for BOH-Cl, respectively.

Figure R22. (a) The calculated density of state of BOH-Br; (b) The calculated partial density of state of bismuth, oxygen and bromine element for BOH-Br, respectively.

Figure R23. (a) The calculated density of state of BOH-I; (b) The calculated partial density of state of bismuth, oxygen and bromine element for BOH-I, respectively.

Comment 12: Line 329-330, ‘The introduction of Γ^- would lead to a more pronounced alteration in the bandgap,’ Does this sentence refer to Table 1?

Reply: “The introduction of Γ^- would lead to a more pronounced extension in the bandgap,” means that the energy band of BOH-I calculated by DOS was smaller than others in the original manuscript, and it was consistent with the result of UV-vis DRS. This phenomenon still exists, after recalculation with new function. Specifically, the band gaps of BOH, BOH-Cl, BOH-Br, and BOH-I calculated by the original method were 2.39 eV, 2.37 eV, 2.33 eV, and 2.20 eV respectively. And the band gaps of BOH, BOH-Cl, BOH-Br, and BOH-I obtained by HSB06 function are 3.17 eV, 3.15 eV, 3.13 eV, and 2.98 eV, separately.

We are sorry that we have omitted the detailed description of Table 1 in the previous manuscript, which may cause misunderstanding among readers. Table 1 shows the contributions of each element to VBM and CBM for BOH and BOH-X (After applying the new calculation function, Table 1 has also been updated, see Table R1 below.). As the energy levels of valence-shell orbitals are different for different halogen atoms, the contributions of these orbitals to the overall band structure are also different. Specifically, as the atomic number of the halide anions goes higher, their contribution to the electron density of VBM becomes larger. In contrast, the halide ions have similar and relatively small contributions (about 0.9%) to CBM, indicating a greater extent of the localization of valence electrons. This leads to a weakened effect of electronic screening and is thus beneficial for carrier separation, as well as the generation and utilization of holes^{R16} (which is consistent with the results discussed in the above catalytic mechanism). The localization of valence electrons and the altered interlayer spacing collectively induce IEF variations between the cation and anion layers.

The above-related content has been supplemented in “**Results: Structure-activity relationship of BOH and BOH-X.**” and the corresponding SI section.

	%	Bi	O	X (Cl/Br/I)
BOH	VB	4.2	95.8	–
	CB	69.2	30.8	–
BOH-Cl	VB	4.5	93.5	2.0
	CB	68.9	30.2	0.9
BOH-Br	VB	4.3	92.2	3.5
	CB	68.8	30.3	0.9
BOH-I	VB	4.3	90.9	4.8
	CB	68.3	30.8	0.9

Table R1| The contributions of Bi, O, and X (X = Cl, Br, I) to the near band edges.

Comment 13: Figure 6. Explicit numbers corresponding to the colors of the legend would be beneficial.

Reply: Thanks for your suggestion. The original manuscript did not properly mark the IEF strength, so the difference in the IEF intensity among samples was not noticeable, which was indeed easy to cause readers confusion. In order to study the role of the IEF more professionally, we recalculated with the HSE06 function. The new calculation results show more pronounced distinctions of the IEF intensity of BOH and BOH-X (Figure R24), which is also consistent with the catalytic activity of the materials (Figure R25). Accordingly, we have updated the calculation content and illustration of the manuscript (changing the original figure 6 to the current figure 5), marked the value in the local IEF in Figure 5 with the corresponding legend color.

Figure R24. DFT calculated the local internal electric field for the four catalysts.

Figure R25. Superimposed plots of benzylamine conversion and calculative IEF intensity for the four catalysts.

Comment 14: In the introduction, it is emphasized that low efficiency of carriers' separation hampers development of sophisticated organic reactions. It was expected that in this work, using the developed semiconductors, authors would give an example of 'sophisticated organic reaction'. However, application of the material is limited to oxidation of benzylamine only, which is rather a trivial reaction.

Reply: Thanks for your comment. It needs to be emphasized that the core point of our manuscript is to introduce an ion-exchange strategy to increase the IEF intensity and then achieve efficient separation and utilization of carriers. Therefore, we chose the model reaction of benzylamine oxidation to verify our design strategy. After proving the strategy's effectiveness, we used BOH-I as a catalyst and designed experiments to oxidize a series of complex imines under the same reaction conditions. The experimental results show that BOH-I is effective for the oxidation of these imines substrates (conversion $\geq 85\%$, selectivity $\geq 93\%$)

(Figure R25). Therefore, it can be seen that the carrier separation efficiency caused by IEF control is greatly improved, and the complex photocatalytic organic reactions can indeed be realized.

The above-related content has been supplemented in “**Results: Photocatalytic performances and mechanism.**” and the corresponding SI section.

Figure R25. Catalytic performance of BOH-I for series of benzylamine derivatives (substrate: 0.1 mmol, catalyst: 20 mg, acetonitrile: 3 mL, visible light irradiation for 6h).

References:

- R1. Kanata-Kito, T. *et al.* Photorefectance characterization of built-in potential in MBE-produced As-grown GaAs surface. *Proc. SPIE* **1286**, 56-66 (1990).
- R2. Im J.S. *et al.* Reduction of oscillator strength due to piezoelectric fields in GaN/Al_xGa_{1-x}N quantum wells. *Phys Rev B* **57**, 9435-9438 (1998).

- R3. Lefebvre P. *et al.* Time-resolved photoluminescence as a probe of internal electric fields in GaN-(GaAl)N quantum wells. *Phys Rev B* **59**: 15363-15367 (1999).
- R4. Morello G. *et al.* Della Sala F, Carbone L *et al.* Intrinsic optical nonlinearity in colloidal seeded grown CdSe/CdS nanostructures: Photoinduced screening of the internal electric field. *Phys Rev B* **78**: 195313 (2008).
- R5. Li, J., Cai, L., Shang, J., Yu, Y. & Zhang, L. Giant enhancement of internal electric field boosting bulk charge separation for photocatalysis. *Adv Mater* **28**, 4059-4064 (2016).
- R6. Chen, X.J., Xu Y., Ma X.G. & Zhu Y.F. Large dipole moment induced efficient bismuth chromate photocatalysts for wide-spectrum driven water oxidation and complete mineralization of pollutants. *National Science Review* **7**, 652–659 (2019).
- R7. Formal, F.L., Sivula K. & Grätzel M. The transient photocurrent and photovoltage behavior of a hematite photoanode under working conditions and the influence of surface treatments. *J Phys Chem C* **116**, 26707-26720 (2012).
- R8. Zhu, Y., Yao, W. & Zong, L. *Photocatalysis: application on environmental purification and green energy*. 524 (2014).
- R9. Pereira, S. *et al.* Splitting of X-ray diffraction and photoluminescence peaks in InGaN/GaN layers. *Materials Science and Engineering: B* **93**, 163-167 (2002).
- R10. Pan, Z., Zhang, G. & Wang, X. Polymeric Carbon Nitride/Reduced Graphene Oxide/Fe₂O₃: All-Solid-State Z-Scheme System for Photocatalytic Overall Water Splitting. *Angew Chem Int Ed Engl* **58**, 7102-7106 (2019).
- R11. Wang, Q. *et al.* Artificial photosynthesis of ethanol using type-II g-C₃N₄/ZnTe heterojunction in photoelectrochemical CO₂ reduction system. *Nano Energy* **60**, 827-835 (2019).
- R12. Yang, M. Q. *et al.* Self-surface charge exfoliation and electrostatically coordinated 2D hetero-layered hybrids. *Nat Commun* **8**, 14224-14224 (2017).

- R13. Wang, Q. *et al.* Single Atomically Anchored Cobalt on Carbon Quantum Dots as Efficient Photocatalysts for Visible Light-Promoted Oxidation Reactions. *Chemistry of Materials* **32**, 734-743 (2020).
- R14. Yu, H. *et al.* Liquid-phase exfoliation into monolayered BiOBr nanosheets for photocatalytic oxidation and reduction. *ACS Sustainable Chemistry & Engineering* **5**, 10499-10508 (2017).
- R15. Nosaka, Y. & Nosaka, A. Y. Generation and detection of reactive oxygen species in photocatalysis. *Chem. Rev* **117**, 11302-11336 (2017).
- R16. Wang, H. *et al.* Giant electron-hole interactions in confined layered structures for molecular oxygen activation. *J Am Chem Soc* **139**, 4737-4742 (2017).

Replies to the Reviewer #2

The manuscript entitled “Anion-Exchanged-Mediated Internal Electric Field: Boosting Photogenerated Carrier Separation and Utilization” presents the synthesis of bismuthoxyhydroxide as nanorods and the effect of halogen substitution of the hydroxides on its photocatalytic abilities. The authors claim that the partial substitution of OH⁻ by halogen atoms enhance the photocatalytic conversion due to the formation of a strong internal electric field. This is an interesting work, but the results and hypothesis are too preliminary for publication in Nature Communication.

Reply: We would like to thank the reviewer for commenting “This is an interesting work.” Based on your suggestions, we have conducted more experiments and changed the computational function to improve this manuscript.

Comment 1: One main reason of the formation of an IEF presented by the author is the decrease of the photoluminescence efficiency. But a decrease of photoluminescence efficiency in solids is more attributed to trap states in the bandgap than an efficient charge carrier dissociation. Thus, this argument cannot be used as it is to prove the existence of an IEF.

Reply: Thanks for pointing it out. The lack of solid IEF evidence is indeed the deficiency of the original manuscript. Therefore, we carried out supplementary experiments to explore the internal electric field intensity for BOH and BOH-X, hoping to provide direct evidence for the existence of IEF. The method of estimate IEF intensity reported by Kanata et al. has been widely adopted, according to Photoreflectance signal amplitude^{R1-R6}. It was based on the effect of internal electric field disturbance on the dielectric constant of the semiconductor so that the photoreflectance signal was generated.

The formula is as follows:

$$F_s = (-2V_s \rho / \epsilon \epsilon_0)^{1/2}$$

Where, F_s is the internal electric field magnitude, V_s is the surface voltage, ρ is the surface

charge density, ϵ is the low-frequency dielectric constant, and ϵ_0 is the permittivity of free space. As ϵ and ϵ_0 are constants, the IEF intensity is determined by the surface voltage (V_s) and the surface charge density (ρ). Therefore, we could qualitatively compare the IEF intensity of BOH and BOH-X according to their $(V_s \rho)^{1/2}$ values.

Based on that, we first applied the open-circuit potentials measurements to evaluate the surface voltages of BOH and BOH-X. As shown in Figure R1, the surface voltage of BOH-I is 0.268 V, which is greater than that of BOH-Br (0.198 V), BOH-Cl (0.175 V), and BOH (0.142 V), indicating the enhanced IEF.

Figure R2. The open-circuit potentials of BOH (a) and BOH-X (b–d).

As Le Formal and Gratzel et al. reported, the accumulated positive charge on the surface is proportional to the integral value, which is calculated from the transient photocurrent density minus the steady-state photocurrent density in the same time^{R7}. Furthermore, the transient photocurrent density measurements were conducted, controlling the same contact

areas with ITO of every sample. The surface charge densities of BOH and BOH-X were then obtained by the integral of the transient anodic photocurrent peaks (Figure R2). As expected, the integral value of the photocurrent response of BOH-I ($148.1 \mu\text{C}\cdot\text{cm}^{-2}$) is the maximum among all specimens, which is twice as magnitude as that of BOH ($70.4 \mu\text{C}\cdot\text{cm}^{-2}$), and also higher than that of BOH-Br ($123.8 \mu\text{C}\cdot\text{cm}^{-2}$) and BOH-Cl ($105.4 \mu\text{C}\cdot\text{cm}^{-2}$).

It can be found that the internal electric field intensity of BOH, BOH-Cl, BOH-Br, and BOH-I gradually increase (Figure R3). Taking the internal electric field intensity of BOH as the “1”, the internal electric field intensity of BOH-I is double BOH’s, while these of BOH-Br and BOH-Cl are “1.6” and “1.4”, respectively.

Figure R2. The transient photocurrent density of BOH (a) and BOH-X (b–d).

Figure R3. The internal electric field intensity of BOH and BOH-X (assuming the IEF intensity of BOH to be “1”).

The enhancement of the IEF intensity endows the photocatalysts with great potential for efficient charge separation and migration. Then, the different carrier separation behaviors of BOH and BOH-X were studied. It is widely known that the transient photocurrent tests (Figure R4) can directly reflect the number of residual electrons and holes after recombination and can qualitatively express the separation efficiency of carriers. The results show that the carrier separation efficiency of BOH is the weakest and that of BOH-Cl, BOH-Br, and BOH-I gradually increase, which is consistent with the results of steady-state fluorescence spectroscopy.

Figure R4. The photocurrent density of BOH and BOH-X.

We assumed the photocatalytic activity of BOH as the “1” and then obtained the catalytic benzylamine activity of BOH, BOH-Cl, BOH-Br, and BOH-I as “1”, “1.6”, “1.8” and “2.0”, separately, based on their catalytic performance of benzylamine oxidation. Similarly, the separation and transport efficiencies of the photocarriers of BOH, BOH-Cl, BOH-Br, and BOH-I are “1”, “1.4”, “1.7”, and “2.0”, respectively, according to their photocurrent response. It can be seen that the variation range of IEF intensity is consistent with the above data for BOH and BOH-X (Figure R5). Therefore, it can be considered that the most crucial reason for the improvement of carrier separation and utilization in BOH-X is the promotion of IEF intensity. Combined with the IEF intensity measurements and the calculation results, we believe that as the atomic number of the introduced halide species goes higher, the ionic radius becomes larger, and the charge distribution between the layers become more uneven; the larger electrostatic potential difference between the layers intensifies the interlayer IEF, and thus promotes the carrier separation and utilization.

The above-related content has been supplemented in “**Results: Synthesis strategy and characterization of BOH nanorods. & Structure-activity relationship of BOH and BOH-X.**” and the corresponding SI section.

Figure R5. Effects of ion exchange on the IEF intensity, carrier separation efficiency, and photocatalytic activity of benzylamine oxidation (assuming all the BOH value to be “1”).

Comment 2: Another argument given to support the idea of an IEF comes from the computations. But these computations are not at the state of the art. The author should look at the literature about the most appropriate computational protocol to characterize a semiconductor for photocatalytic application. They should use the range separated hybrid functional HSE06 for their calculation and deeply characterize their materials (bandgap, dielectric constant, effective masses). Furthermore, the Figure 6 is absolutely unclear. What information do we extract from the charge density? Figure 6f is too small to see the differences. From the Table 1, it appears that the DOS is almost unchanged upon halogen substitution. How do the authors explain the apparition of an IEF? Clearly, the DFT part could be much stronger and, if placed earlier in the paper, could support more the conclusion.

Reply: We thank the reviewer for the valuable suggestion. In order to make the theoretical calculation more in line with the actual situation, we adopted a more suitable range separated hybrid functional HSE06 to describe the semiconductor for photocatalytic application, according to your suggestion. Compared with the PBE function, the band gaps obtained by HSE06 function increase, better aligning with the experimental results. Therefore, it can be seen that the PBE function underestimates the energy band of the materials, while the HSE06 function is more suitable for the calculation of photocatalytic materials. The calculation result with HSE06 function further demonstrates that "the introduction of halogen ions enhances the IEF intensity " in this manuscript.

First, using HSE06 function to calculate ^{R8-R9}, we found that DOS changed more reasonably for BOH-X (Figure R6–9). Specifically, the band gaps of BOH, BOH-Cl, BOH-Br, and BOH-I calculated by the original method were 2.39 eV, 2.37 eV, 2.33 eV, and 2.20 eV respectively, which obtained by HSE06 function are 3.17 eV, 3.15 eV, 3.13 eV, and 2.98 eV, separately. Taking into account that our theoretical calculation model is established with an ion exchange capacity of 20 % (the actual ion exchange capacity is approximately 10%, the models of introducing more halogen ions are adopted to reduce the calculation cost and make the effect of halogen ions more intuitive.), the new calculation results are more align with the UV-Vis DRS data.

Figure R6. (a) The calculated density of state of BOH; (b) and (c) The calculated partial density of state of bismuth and oxygen element for BOH, respectively.

Figure R7. (a) The calculated density of state of BOH-Cl; (b) The calculated partial density of state of bismuth, oxygen and chlorine element for BOH-Cl, respectively.

Figure R8. (a) The calculated density of state of BOH-Br; (b) The calculated partial density of state of bismuth, oxygen and bromine element for BOH-Br, respectively.

Figure R9. (a) The calculated density of state of BOH-I; (b) The calculated partial density of state of bismuth, oxygen and iodine element for BOH-I, respectively.

At the same time, the new calculation results show that the role of halogen ions is also more prominent. As the energy levels of valence-shell orbitals are different for different halogen atoms, the contributions of these orbitals to the overall band structure are also different. Specifically, as the atomic number of the halide anions goes higher, their contribution to the electron density of VBM becomes more significant, in contrast, the halide ions have a similar and relatively small contribution (about 0.9%) to CBM, indicating a greater extent of the localization of valence electrons. This leads to a weakened effect of electronic screening and is thus beneficial for the separation of electrons and holes and the generation and utilization of holes^{R10} (which is consistent with the results discussed in the above catalytic mechanism). The localization of valence electrons and the altered interlayer spacing collectively induce a change

in the IEF between the cation and anion layers.

	%	Bi	O	X (Cl/Br/I)
BOH	VB	4.2	95.8	–
	CB	69.2	30.8	–
BOH-Cl	VB	4.5	93.5	2.0
	CB	68.9	30.2	0.9
BOH-Br	VB	4.3	92.2	3.5
	CB	68.8	30.3	0.9
BOH-I	VB	4.3	90.9	4.8
	CB	68.3	30.8	0.9

Table R1| The contributions of Bi, O, and X (X = Cl, Br, I) to the near band edges.

After calculating HSE06 functional, the IEF intensity of BOH and BOH-X changed more obviously (Figure R10). In particular, according to the initial calculation results, the local electric field intensity of each sample is 0.42 eV (for BOH), 0.47 eV (for BOH-Cl), 0.49 eV (for BOH-Br), and 0.51 eV (for BOH-I). In comparison, the local IEF intensity calculated with HSE06 function is 10.8 eV, 11.6 eV, 11.9 eV, and 12.3 eV (corresponding to BOH, BOH-Cl, BOH-Br, and BOH-I, respectively), which are consistent with the results of the IEF intensity measurement experiments (see Comments 1), as well as the catalytic activity of BOH and BOH-X (Figure R11). The results prove that the intrinsic driving force of BOH-X performance improvement is the increase of IEF intensity caused by the introduction of halogen ions.

The above-related content has been supplemented in “**Results: Structure-activity relationship of BOH and BOH-X.**” and the corresponding SI section.

Figure R10. DFT calculated the local internal electric field for the four catalysts.

Figure R11. Superimposed plots of benzylamine conversion and calculative IEF intensity over the samples.

Comment 3: Figure 4d, the authors should give the band positions with respect to an electrode reference.

Reply: We thank the reviewer for the helpful suggestions. We have redesigned the experiments to obtain more accurate band structures concerning an electrode reference of BOH and BOH-X. As Figure R12 shown, all BOH-X and BOH samples have a similar conduction band minimum (CBM) (at -1.83--1.93 eV), but for Valence Band Maxime (VBM), VBM of BOH is similar to that of BOH-Cl and BOH-Br (about 1.70 eV) while BOH-I has the highest VBM (1.24 eV). In summary, the anion exchange only induces a moderate change in VMB only for BOH-I, whereas its influences on the band structures of BOH-X are somewhat limited. The specific methods are as follows:

First, we measured the XPS valence band spectrum of each catalyst (Figure R13(a-c)) and obtained the distances from the VBM to the Fermi level as 1.92 eV, 1.90 eV, 1.88 eV, and 1.62 eV corresponding to BOH, BOH-Cl, BOH-Br, and BOH-I respectively. According to the Tauc plot (Figure R13(d)) obtained by UV-vis DRS conversion, the band gaps of BOH, BOH-Cl, BOH-Br, and BOH-I are 3.60 eV, 3.56 eV, 3.55 eV, and 3.13 eV separately. Correspondingly, the distances from the CBM to the Fermi level are calculated to be 1.68 eV, 1.66 eV, 1.67 eV, and 1.51 eV for BOH, BOH-Cl, BOH-Br, and BOH-I, respectively (using the equation $E_{CB} = E_g - E_{VB}$).

In order to assess the Fermi levels for BOH and BOH-X, we conducted the Mott-Schottky measurements at the potential of -0.1 V to -0.7 V (at a fixed frequency of 500 Hz). Based on the Mott-Schottky formula, $C_{sc}^{-1} = 2(\Delta\Phi_{sc} - RT/F) (q\epsilon\epsilon_0N)^{-1}$ (where $\Delta\Phi_{sc} = V - V_{fb}$, V_{fb} is the flat band potential, T is the Kelvin temperature, F is the Faraday constant, R is the gas constant, ϵ and ϵ_0 are the semiconductor dielectric constant and vacuum dielectric constant, respectively, q is the charge quantity, and N is the doping concentration), we plotted C_{sc}^{-1} versus V (Figure S22), and then obtained the flat bands of BOH, BOH-Cl, BOH-Br and BOH-I through the intercept of the abscissa $V_0 = V_{fb} + RT/F$. Therefore, the potentials of BOH, BOH-Cl, BOH-Br and BOH-I are 0.15 eV, 0.21 eV, 0.03 eV and 0.02 eV, (versus Ag/AgCl at pH 6.80) respectively,

which are -0.25 eV, -0.19 eV, -0.17 eV and -0.38 eV separately, relative to the normal hydrogen electrode at pH=0 (NHE). Since the Fermi level E_f and V_{fb} have the same value, combined with the valence band spectrum and UV-vis DRS test results, the energy band diagrams of BOH and BOH-X relative to the normal hydrogen electrode are finally obtained as Figure R12.

The above-related content has been supplemented in “**Results: Structure-activity relationship of BOH and BOH-X.**” and the corresponding SI section.

Figure R12. The schematic diagram of band structure of BOH and BOH-X.

Figure R13. The characterization of BOH and BOH-X: (a–c) valence band spectra; (d) Tauc plots.

Comment 4: The authors should provide the response spectra as a function of the wavelength to see if the increase of efficiency of the BOH-I is not only due to a redshift of the absorption spectra.

Reply: Thanks for your valuable suggestion. In order to explore the redshift effect of BOH-I, we designed photocatalytic experiments with monochromatic light of different wavelengths.

According to the Tauc plot (Figure R13(d)) and the formula $\lambda_g = 1240/E_g$ (λ_g is the light absorption threshold of the material; E_g is the band gap.), the light absorption threshold values

of BOH, BOH-Cl, BOH-Br, and BOH-I are 344 nm, 348 nm, 350 nm, and 396 nm, respectively. Therefore, the redshift region of BOH-I is mainly concentrated in the range of 340 nm–400 nm. Based on this, the monochromatic lights with four wavelengths (365 nm, 405 nm, 450 nm, and 500 nm) were selected (Figure R14). The results show that whether the wavelength of the light source is in the BOH-I redshift region or not, the photoactivity still follows the previous order (BOH < BOH-Cl < BOH-Br < BOH-I), which proves that the redshift of the absorption spectrum is not an essential advantage for BOH-I. Combined with the qualitative IEF intensity measurements, we can attribute the high activity of BOH-I mainly to its more vigorous IEF intensity.

The above-related content has been supplemented in “**Results: Photocatalytic performances and mechanism.**” and the corresponding SI section.

Figure R14. Photocatalytic activity over BOH and BOH-X by introducing different single wavelength light.

Comment 5: On figure 5a, the nitrile group on the side product is linear. Please correct the structure and the TOC that also contains this structure.

Reply: We thank the reviewer for the careful examination of our paper. We have re-examined our manuscript and the supporting information and made some revisions to the wording and grammar. The revised parts are listed below.

Corresponding changes:

1. Page 2, Line 7, and Page 4, Line 9: " basic bismuth nitrate compound" was corrected to " bismuth oxyhydroxide compound".
2. Page 6, Line 10: "NRs" was corrected to "nanorods".
3. Page 6, Line 17: "with" was added.
4. Page 11, Line 17: "1 eV" was corrected to "0.2 eV".
5. Page 15, Line 7: "Imide" was corrected to "Imine".
6. Page 26: the by-products, including benzaldehyde and benzonitrile, were added to TOC (Scheme 1).
7. Page 34, Line 3: "catalytic" was corrected to "Catalytic".
8. Page 34, Line 4: " results of" was corrected to " Results from".
9. Page 34, Line 4: "catalytic" was corrected to "Catalytic".
10. Page 34: the structural formula of benzonitrile in Figure 5(a) was corrected.
11. Page 40, Line 17: " stirred" was corrected to " stirring".
12. Page 43, Line 20: " of pro-posed" was corrected to " proposed ".

References:

- R1. Kanata-Kito, T. *et al.* Photorefectance characterization of built-in potential in MBE-produced As-grown GaAs surface. *Proc. SPIE* **1286**, 56-66 (1990).
- R2. Im J.S. *et al.* Reduction of oscillator strength due to piezoelectric fields in GaN/Al_xGa_{1-x}N quantum wells. *Phys Rev B* **57**, 9435-9438 (1998).
- R3. Lefebvre P. *et al.* Time-resolved photoluminescence as a probe of internal electric fields in GaN-(GaAl) N quantum wells. *Phys Rev B* **59**: 15363-15367 (1999).

- R4. Morello G. et al. Della Sala F, Carbone L *et al.* Intrinsic optical nonlinearity in colloidal seeded grown CdSe/CdS nanostructures: Photoinduced screening of the internal electric field. *Phys Rev B* **78**: 195313 (2008).
- R5. Li, J., Cai, L., Shang, J., Yu, Y. & Zhang, L. Giant enhancement of internal electric field boosting bulk charge separation for photocatalysis. *Adv Mater* **28**, 4059-4064 (2016).
- R6. Chen, X.J., Xu Y., Ma X.G. & Zhu Y.F. Large dipole moment induced efficient bismuth chromate photocatalysts for wide-spectrum driven water oxidation and complete mineralization of pollutants. *National Science Review* **7**, 652–659 (2019).
- R7. Formal, F.L., Sivula K. & Grätzel M. The transient photocurrent and photovoltage behavior of a hematite photoanode under working conditions and the influence of surface treatments. *J Phys Chem C* **116**, 26707-26720 (2012).
- R8. Heyd J.; Scuseria G. E. & Ernzerhof M. Hybrid functionals based on a screened Coulomb potential. *J Chem Phys.* **118**, 8207–8215 (2003).
- R9. Heyd J.; Scuseria G. E. & Ernzerhof M. Erratum: “Hybrid functionals based on a screened Coulomb potential” [J. Chem. Phys. 118, 8207 (2003)] *J Chem Phys* **124**, 219906 (2006).
- R10. Wang, H. *et al.* Giant electron-hole interactions in confined layered structures for molecular oxygen activation. *J Am Chem Soc* **139**, 4737-4742 (2017).

REVIEWERS' COMMENTS

Reviewer #1 (Remarks to the Author):

In this version of the manuscript, authors responsibly addressed most of the comments. They added plenty of experimental data and adjusted discussion. I enjoyed reading a revised version of the manuscript. I recommend acceptance of the manuscript for publication in Nature Communications after following changes are implemented.

1. Authors synthesized more materials with progressively increasing content of halide ions using ion exchange reaction. In the footnotes to the figures, such as Figure 3a-d, S21, S22, S23, etc., it should be specified what halide content in BOH-X is implied.
2. Page 14, line 22, 'from the reduction of electrons by oxygen'. 'from the reduction of oxygen by electrons'.
3. In this version of the manuscript authors removed transient photoluminescence spectra, but description of the method is given in the corresponding section. I think this is a remaining from the previous version of the manuscript.
4. Description of the method used for theoretical calculations (page 34, line 2). It is not clear which 'sulfides' authors refer to. All materials prepared in this work do not contain sulfur.
5. Page 32, line 20. 'ESR spectra were recorded.'

Reviewer #2 (Remarks to the Author):

The answers provided by the authors to all the remarks are convincing. The conclusions are now perfectly supported by data and this is a very interesting work. I recommend the publication in Nature Communications.

Point-by-point response

Replies to the Reviewer #1

In this version of the manuscript, authors responsibly addressed most of the comments. They added plenty of experimental data and adjusted discussion. I enjoyed reading a revised version of the manuscript. I recommend acceptance of the manuscript for publication in Nature Communications after following changes are implemented.

Reply: We appreciate for your positive evaluation to our work. Following your advices and suggestions, we carefully modified the manuscript and supplementary information accordingly.

Comment 1: Authors synthesized more materials with progressively increasing content of halide ions using ion exchange reaction. In the footnotes to the figures, such as Figure 3a-d, S21, S22, S23, etc., it should be specified what halide content in BOH-X is implied.

Reply: Thanks for pointing it out. To avoid misunderstanding, we have supplemented “and, unless otherwise stated, BOH-X samples refer to ion exchange products with 0.1 mmol halogenated potassium added” in Page 11, Line 5–6. And we also corrected the footnotes to the figures in Supplementary Figure 21 and 22, and so on.

Comment 2: Page 14, line 22, ‘from the reduction of electrons by oxygen’. ‘from the reduction of oxygen by electrons’.

Reply: We thank the reviewer for the careful examination of our paper. In Page 15, Line 9, "from the reduction of electrons by oxygen" was corrected to "from the reduction of oxygen by electrons".

Comment 3: In this version of the manuscript authors removed transient photoluminescence spectra, but description of the method is given in the corresponding

section. I think this is a remaining from the previous version of the manuscript.

Reply: Thanks for pointing it out. We have deleted the extraneous words in “**Methods: Characterizations.**”.

Comment 4: Description of the method used for theoretical calculations (page 34, line 2). It is not clear which ‘sulfides’ authors refer to. All materials prepared in this work do not contain sulfur.

Reply: Thanks for your careful examination. We have corrected "the sulfides" to "BOH and BOH-X" in Page 24, Line 3.

Comment 5: Page 32, line 20. ‘ESR spectra were recorded.’

Reply: Thanks for your comment, we have revised this error in Page 22, Line 21, accordingly.

Replies to the Reviewer #2

The answers provided by the authors to all the remarks are convincing. The conclusions are now perfectly supported by data and this is a very interesting work. I recommend the publication in Nature Communications.

Reply: We really appreciate your positive feedback for our revised manuscript!

Finally, on behalf of all authors, we would like to thank the Editor and Reviewers for your precious time and helpful suggestions for improving the quality of this manuscript and keeping the high standard of the *Nature Communications*.

Best regards

Yadong Li